# Towards Better Instruction Following Retrieval Models

## Abstract

Modern information retrieval (IR) models, trained exclusively on standard `<query, passage>` pairs, struggle to effectively interpret and follow explicit user instructions. We introduce `InF-IR`, a large-scale, high-quality training corpus tailored for enhancing retrieval models in Instruction-Following IR. InF-IR expands traditional training pairs into over 38,000 expressive `<instruction,query,passage>` triplets as *positive* samples. In particular, for each positive triplet, we generate two additional hard *negative* examples by poisoning both instructions and queries, then rigorously validated by an advanced reasoning model (`o3-mini`) to ensure semantic plausibility while maintaining instructional incorrectness. Unlike existing corpora that primarily support computationally intensive reranking tasks, the highly contrastive positive-negative triplets in `InF-IR` further enable efficient representation learning to facilitate direct embedding-based retrieval. Using this corpus, we train `InF-Embed`, an instruction-aware Embedding model optimized through contrastive learning and instruction-query attention mechanisms to align retrieval outcomes precisely with user intents. Extensive experiments across multiple instruction-based retrieval benchmarks demonstrate that `InF-Embed` significantly improves the instruction-following capability for both embedding-based (+9.0 p-MRR) and auto-regressive language models (+4.2 p-MRR) across different model sizes.

## 1 Introduction

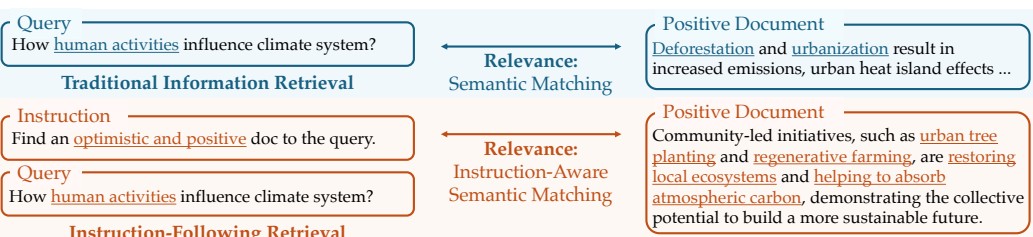

Figure 1: Example of original information retrieval compared to instruction-following retrieval.

Information retrieval (IR) systems play an important role in efficiently accessing relevant information from vast document collections (Robertson et al., 1995; Karpukhin et al., 2020). Despite notable advancements, conventional retrieval models often struggle to accurately interpret and align with specific user requests, retrieving information based primarily on lexical or semantic matching while overlooking nuanced intents explicitly expressed in complex user queries (Figure 1). Modern language models (LMs) serving as the backbones of retrieval systems has demonstrated strong potential to incorporate instruction-following capabilities (Ouyang et al., 2022; Wang et al., 2023b), enabling retrievers to understand and respond accurately to a diverse set of user requests (Su et al., 2023; Asai et al., 2023; Jiang et al., 2024). Instruction-following IR has emerged as an effective paradigm for explicitly guiding retrieval systems through detailed user instructions (Weller et al., 2024; 2025; Muennighoff et al., 2024), thereby enhancing retrieval accuracy and user satisfaction.

In a standard instruction-following IR framework, detailed user instructions are incorporated alongside queries to condition the retrieval process (Weller et al., 2024; Oh et al., 2024). However, embedding

models typically struggle with effectively interpreting and following detailed instructions; conversely, modern decoder-only LMs inherently lack robust representation learning capabilities, inadequately capturing the complex interactions among instructions, queries, and documents (Xiao et al., 2024; Wang et al., 2022a; Izacard et al., 2021). This fundamental *dilemma* underscores the pressing need for an effective embedding-based instruction-aware retrieval model that simultaneously excels in both efficiently encoding and accurately interpreting complex *instruction-query-passage* interactions. Addressing this challenge requires high-quality training resources specifically tailored for instruction-aware representation learning; unfortunately, existing instruction-following IR datasets (Petroni et al., 2021; Thakur et al., 2021; Muennighoff et al., 2023; Oh et al., 2024; Zhou et al., 2025; Sun et al., 2024; Su et al., 2024) serve primarily as evaluation benchmarks with insufficient training data.

Recent studies (Weller et al., 2024; 2025) employ large language models (LLMs) to synthesize both relevant and irrelevant documents corresponding to specific *instruction-query* pairs. Yet, they often rely merely on binary relevance signals or simplified negative examples, failing to capture the intricate relational dynamics inherent in instruction-based retrieval tasks. Moreover, current training paradigms focus heavily on computationally intensive reranking tasks with decoder-only architectures (Weller et al., 2024), thereby neglecting the substantial efficiency and scalability advantages of embedding-based retrieval models. To summarize, it is still crucial yet challenging to effectively and efficiently unleash the capability of retrieval models for complex instruction-following IR.

In this study, we introduce `InF-IR`, a large-scale training corpus designed to advance instruction-following capabilities in retrieval models. We extend traditional retriever training samples by transforming standard `<query,passage>` pairs into expressive `<instruction,query,passage>` triplets, explicitly modeling complex interactions in instruction-following IR. Specifically, we generate diverse instruction-query combinations paired with corresponding retrieved documents as positive samples, while systematically poisoning both instructions and queries separately to create challenging negative samples. To further strengthen representation learning, we employ an advanced reasoning model (`o3-mini`) to ensure negative sample quality by validating semantic plausibility while maintaining instructional misalignment. The resulting `InF-IR` comprises 38,759 *positive* samples and 77,518 meticulously crafted hard *negative* samples, effectively guiding retrievers to accurately interpret user intentions while distinguishing between semantically similar but instructionally distinct contexts. Importantly, `InF-IR` not only supports training large, computationally expensive auto-regressive LMs, but also enables efficient training and scaling of smaller embedding-based models for instruction-aware representation learning. Building upon `InF-IR`, we propose `InF-Embed`, an instruction-aware text embedding model trained via contrastive learning and instruction-query attention to optimize embeddings, accurately capturing complex relationships among instructions, queries, and retrieved documents. Our key contributions can be summarized as follows:

- (i) *Dataset Wise*, we introduce `InF-IR`, a publicly available *large-scale, high-quality training corpus* specifically designed to enhance retrieval models in instruction-following IR. `InF-IR` features over 38,000 expressive `<instruction,query,passage>` triplets with carefully crafted hard negative examples, effectively addressing the critical shortage of high-quality training resources for instruction-aware representation learning;

- (ii) *Methodology Wise*, we propose `InF-Embed`, *an instruction-aware embedding model* optimized via contrastive learning and instruction-query attention. `InF-Embed` efficiently encodes and precisely interprets complex user instructions, resolving the efficiency-effectiveness trade-off faced by traditional decoder-only and encoder-only instruction-following retrieval models; and

- (iii) *Experimental and Benchmark Wise*, extensive empirical evaluations demonstrate that `InF-Embed` consistently improves instruction-following performance for both embedding-based (+9.0 p-MRR) and auto-regressive (+4.2 p-MRR) LMs, facilitated by our diverse training corpus, `InF-IR`. Moreover, we systematically benchmark a comprehensive suite of contrastive learning objectives across multiple embedding models and LMs with varying sizes, thereby supporting rapid future advances in instruction-following retrieval systems.

## 2 RELATED WORKS

**Instruction-Following Retrieval Datasets.** Integrating explicit instructions into IR models represents a recent research focus that contrasts with traditional dense retrievers emphasizing phrase-level semantic matching (Wang et al., 2022a; Izacard et al., 2021). While several datasets (Petroni

Table 1: Summary of existing instruction-following IR datasets. "I", "Q", and "P" denote "instruction", "query", and "passage", respectively. "$-$" denotes negative samples; for example, "I$^-$" indicates contrasting instruction for negative sample generation. Notations are consistent across tables.

| Datasets | Eval. | Train | (Q, P)$^+$ | I$^+$ | I$^-$ | Q$^-$ | P$^-$ | Quality Check | #I | #Q | #P | Avg. \|I\| | Avg. \|Q\| | Avg. \|P\| |
|---|---|---|---|---|---|---|---|---|---|---|---|---|---|---|
| KILT (2021) | ✓ | ✗ | ✓ | ✗ | ✗ | ✗ | ✗ | - | - | 50.7K | 5.9M | - | 160.83 | 18.23 |
| BEIR (2021) | ✓ | ✗ | ✓ | ✗ | ✗ | ✗ | ✗ | - | - | 54.3K | 52.8M | - | 14.78 | 113.77 |
| MTEB (2023) | ✓ | ✗ | ✓ | ✗ | ✗ | ✗ | ✗ | - | - | 1.0M | 172M | - | 25.64 | 100.14 |
| InstructIR (2024) | ✓ | ✗ | ✓ | ✓ | ✗ | ✗ | ✗ | gpt-4 | 9.9K | 9.9K | 16.1K | 49.04 | 5.57 | 91.23 |
| FollowIR (2024) | ✓ | ✗ | ✓ | ✓ | ✗ | ✗ | ✗ | gpt-4 | 104 | 104 | 98.3K | 43.51 | 11.44 | 122.69 |
| Bright (2024) | ✓ | ✗ | ✓ | ✗ | ✗ | ✗ | ✗ | - | - | 1.3K | 1.3M | - | 203.05 | 343.01 |
| MAIR (2024) | ✓ | ✗ | ✓ | ✓ | ✗ | ✗ | ✗ | - | 805 | 10.0K | 4.3M | 33.18 | 315.16 | 547.51 |
| InfoSearch (2025) | ✓ | ✗ | ✓ | ✓ | ✗ | ✗ | ✗ | gpt-4 | 1.6K | 600 | 6.4K | 17.21 | 8.19 | 175.98 |
| IFIR (2025) | ✓ | ✗ | ✓ | ✓ | ✗ | ✗ | ✗ | gpt-4o | 2.1K | 943 | 1.4M | 99.35 | 36.52 | 224.97 |
| Promptriever (2025) | ✓ | ✓ | ✓ | ✓ | ✗ | ✗ | ✓ | FollowIR-7B | 489K | 489K | 1.6M | 103.2 | 5.95 | 56.27 |
| **InF-IR (Ours)** | ✓ | ✓ | ✓ | ✓ | ✓ | ✓ | ✓ | o3-mini | 77.5K | 77.5K | 116.2K | 35.57 | 8.06 | 55.2 |

et al., 2021; Thakur et al., 2021; Oh et al., 2024; Su et al., 2024; Sun et al., 2024; Zhou et al., 2025; Muennighoff et al., 2023; Weller et al., 2024; 2025; Song et al., 2025) have emerged to comprehensively evaluate the instruction-following capabilities of retrieval models, there remains a notable scarcity of sufficient and high-quality training resources (Table 1). FollowIR (Weller et al., 2024) offers a small set of 104 instructions with simple binary relevance signals. Although Promptriever (Weller et al., 2025) contributes a significantly larger training set, it generates negative examples by only contrasting documents and relies extensively on an under-trained small instruction-tuned LM for quality assurance. Motivated by these limitations, we introduce `InF-IR`, an instruction-following IR data synthesis pipeline that systematically generates challenging negative examples by jointly contrasting instructions, queries, and documents. Moreover, `InF-IR` incorporates rigorous quality validation, resulting in a high-quality corpus of representative positive-negative triplets specifically designed to enhance instruction-aware contrastive learning.

**Instruction-Following Retrieval Models.** LMs as backbones of information retrievers enable ad-hoc search systems to retrieve with user instructions when responding to complex queries (Wang et al., 2023a; Moreira et al., 2024; Su et al., 2023; Asai et al., 2023). Early attempts to incorporate instructions into retrieval systems have often relied on decoder-only LLMs, formulating the retrieval task as a specialized text generation or reranking problem. For example, FollowIR (Weller et al., 2024) fine-tunes a LM as a reranker, achieving notably better alignment with user instructions than standard bi-encoder retrievers. Additionally, GritLM (Muennighoff et al., 2024) integrates representation and generative instruction tuning into a unified decoder-style architecture, capable of handling both generative and embedding tasks simultaneously by distinguishing them through instructions. Promptriever (Weller et al., 2025) fine-tunes RepLLaMA upon query-level instruction data to improve retrieval efficiency and adaptability to diverse query instructions. In contrast to existing instruction-following IR models that rely on powerful yet inefficient and less scalable auto-regressive LMs, we hypothesize that embedding models as retrieval backbones can effectively address diverse user requests through advanced instruction-aware representation learning.

## 3 PRELIMINARIES

**Noise Contrastive Estimation.** We begin by formulating a ranking-based noise contrastive estimation (NCE) objective (Ma & Collins, 2018; Gutmann & Hyvärinen, 2010; Henderson et al., 2017; Yang et al., 2019) from a conditional modeling perspective. Specifically, consider a model that estimates a conditional distribution $\mathbb{P}(\mathbf{y} \mid \mathbf{x})$, where $\mathbf{x}$ and $\mathbf{y}$ represent arbitrary combinations of target variables. We define a scoring function $s_\theta(\mathbf{x}, \mathbf{y})$ parameterized by learnable parameters $\theta$, quantifying the relevance between a given pair $(\mathbf{x}, \mathbf{y})$. Given a training set $\mathcal{D} = \{\mathbf{x}_i, \mathbf{y}_i\}_{i=1}^n$ and an arbitrary minibatch $\mathcal{B} \subseteq \mathcal{D}$[1] sampled during training, we introduce a predefined negative sampling distribution $\mathbb{P}_\mathcal{B}^-(\cdot)$ for generating negative examples within each minibatch. The resulting NCE objective using in-batch negatives is formulated as follows:

$$\ell_{\text{NCE}}(\theta) = -\mathbb{E}_{i \sim \mathcal{B}} \left[ \log \frac{\exp\left(s_\theta\left(\mathbf{x}_i, \mathbf{y}_i\right)\right)}{\sum_{\mathbf{y}_k \sim \mathbb{P}_\mathcal{B}^-(\mathbf{y})} \exp\left(s_\theta\left(\mathbf{x}_i, \mathbf{y}_k\right)\right)} \right]. \tag{1}$$

---

[1]For simplicity, $\mathcal{D}$ and $\mathcal{B}$ also represent the sets of indices corresponding to the sample pairs they contain.

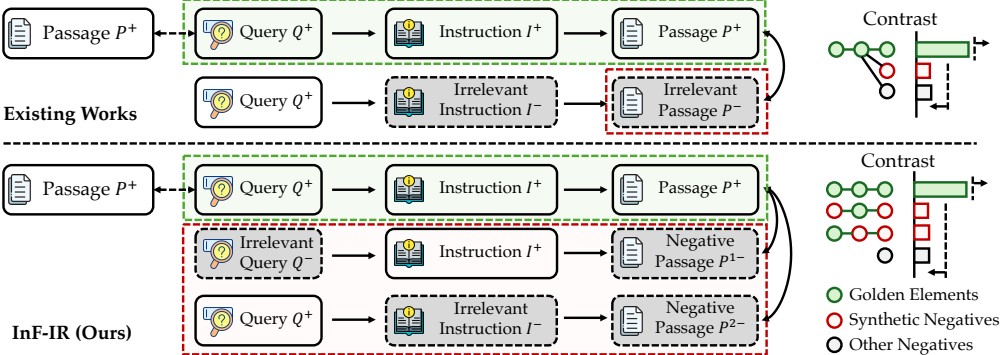

Figure 2: Hard negative samples in `InF-IR` generated by poisoning both instructions and queries.

**Dense Passage Retrieval with Instructions.** Consider a corpus $\mathcal{P} = \{P_i\}_{i=1}^N$ comprising a large set of candidate retrieval passages. Given a query $Q$ paired with an instruction $I$ provided by the user, an instruction-following retrieval model aims to retrieve a concise subset of passages from $\mathcal{P}$ that best satisfies the instruction and query. We denote this targeted positive subset as $\mathcal{P}^+ = \{P_j^+\}_{j=1}^M$, where $M \ll N$, and correspondingly define the negative set as $\mathcal{P}^- = \mathcal{P} \setminus \mathcal{P}^+$. During training, we adopt the NCE to approximate the conditional distribution $\mathbb{P}(P^+|I, Q)$ for the retriever model parameterized by $\theta$, initiating by defining $\mathbf{x} = P$ and $\mathbf{y} = (I, Q)$. Specifically, the objective encourages aligning representations of matching instruction-query-passage triplets $(P^+, I, Q)$, while simultaneously promoting separation of representations corresponding to non-matching triplets $(P^-, I, Q)$. In the retrieval phase, the learned scoring function $s_\theta(P, I, Q)$ quantifies the similarity between candidate passages $P$ and instruction-query pairs $(I, Q)$. Instructions $I$ provide essential supplementary context, specifying various retrieval dimensions such as formatting, stylistic preferences, passage length, or user-specific details such as background knowledge or profiles (Weller et al., 2024; Wang et al., 2022b; Oh et al., 2024). By incorporating instructions, the retrieval model flexibly adapts to diverse user intents, thereby enhancing personalization and utility of retrieved passages $\mathcal{P}^+$.

## 4 `InF-IR`: INSTRUCTION-FOLLOWING IR TRAINING CORPUS

### 4.1 DATA CURATION

In this section, we present `InF-IR`, a large-scale training corpus specifically curated for training a bi-encoder retrieval model capable of effectively following instructions. To ensure generalizability, we utilize MS MARCO (Bajaj et al., 2018) as our seed dataset to construct corresponding `<instruction, query, passage>` tuples.[2] MS MARCO provides a large-scale, general-domain dataset consisting of anonymized real-world queries paired with human-annotated relevant passages. We selected MS MARCO because of its extensive query-passage coverage and high-quality annotations, which provide a solid foundation, allowing us to focus primarily on instruction generation.

**Overview.** Our data curation pipeline proceeds in three stages: (i) We first synthesize explicit instructions aligned to each query-passage pair, creating positive tuples `<instruction, query, passage>`; (ii) To enhance discriminative representation learning, we then employ `gpt-4o-mini` (Hurst et al., 2024) to generate challenging negative examples by introducing subtle alterations to instructions and queries; and (iii) We rigorously validate tuple quality using `o3-mini` as a proxy evaluator, filtering out low-quality tuples where the intended passage relevance is ambiguous or not clearly identifiable.

**Instruction Generation.** We initiate data synthesis by generating a suitable instruction for each query-passage pair in MS MARCO. We prompt `gpt-4o-mini` to produce instructions that add specificity or stylistic context, thereby explicitly linking queries more precisely to their corresponding ground-truth passages. Leveraging `gpt-4o-mini` enables scalable instruction generation with a careful balance between effectiveness and computational efficiency.

**Contrastive Negatives.** To facilitate effective representation learning, we generate challenging negative samples by systematically altering instructions and queries independently, forcing the retrieval model to distinguish subtle differences in relevance. Unlike traditional retrieval setups relying solely on `<query, passage>` pairs, instruction-following retrieval introduces an additional

---

[2]We use the MS MARCO v2.1 available at https://huggingface.co/datasets/microsoft/ms_marco.

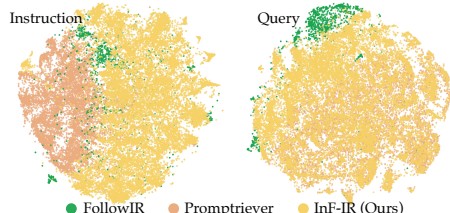
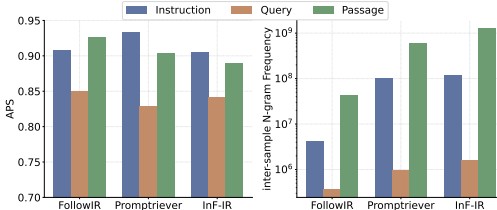

(a) t-SNE Visualization of Semantic Coverage

(b) Diversity Metrics, APS ($\downarrow$) and INGF ($\uparrow$)

Figure 4: Visualization and diversity analysis of synthetic training samples from `InF-IR`.

dimension, the `instruction`. Thus, effective negative sampling must fulfill two criteria: (1) negative samples must be sufficiently different from positives to alter tuple relevance significantly; and (2) they should retain close semantic similarity to positives, enabling models to detect nuanced differences.

To fulfill these criteria, we instruct `gpt-4o-mini` to subtly alter (*i.e.*, poison) the original positive instruction $I^+$ and query $Q^+$, producing closely related yet instructionally misaligned negatives $I^-$ and $Q^-$. By combining the negatively modified instruction with the original query $(I^-, Q^+)$ and vice versa $(I^+, Q^-)$, we obtain two new negative passages $P^{1-}$ and $P^{2-}$. By contrasting these carefully generated negative tuples $(I^-, Q^+, P^{1-})$ and $(I^+, Q^-, P^{2-})$ against the original positive tuple $(I^+, Q^+, P^+)$, our training encourages the model to more accurately capture subtle variations in instruction, query, and passage relevance (Figure 2). Additional details are available in section B.

**Data Quality Check.** To ensure the quality and semantic consistency of our synthetic data, we employ an advanced reasoning model, `o3-mini`, for quality evaluation. This validation procedure rigorously verifies whether the generated instructions preserve the original positive relevance of <query,passage> pairs from MS MARCO. We specifically check consistency across all combinations: the positive tuple $(I^+, Q^+, P^+)$, and the two negative variations $(I^-, Q^+, P^{1-})$ and $(I^+, Q^-, P^{2-})$. For each validation scenario, we simulate the instruction retrieval task by presenting `o3-mini` with the instruction and query alongside the positive passage, closely-related negatives, and additional distractive passages randomly sampled from MS MARCO. We then prompt `o3-mini` to identify the most relevant passage. Only tuples that yield consistent and unambiguous relevance judgments across all three scenarios are retained, while others are discarded. This rigorous filtering maintains high quality data and ensures reliable model training.

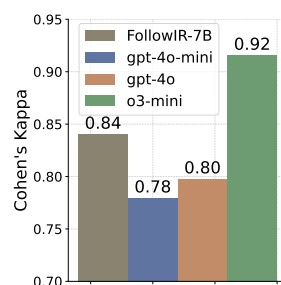

Figure 3: Cohen's kappa from 100 random samples.

To validate the effectiveness and reliability of our quality-check procedure, we conducted a human annotation study. Annotators were asked to identify the most relevant passage for a given instruction-query pair from a set of distractors, including generated negatives from above as well as in-batch negatives. Figure 3 reports the average agreement scores (Cohen's Kappa) between human annotators and various models. The results clearly indicate that `o3-mini` achieves higher agreement with human judgments compared to other models, including `FollowIR-7B`, `gpt-4o-mini`, and `gpt-4o`, thus confirming the robustness and validity of our filtering strategy.

## 4.2 DATA VISUALIZATION AND ANALYSIS

We conduct a comparative experimental analysis using 10,000 random samples from each of FollowIR (Weller et al., 2024), Promptriever (Weller et al., 2025), and our `InF-IR`.

**Qualitative Analysis of Semantic Coverage.** To qualitatively assess topic coverage of our generated training data compared to FollowIR (Weller et al., 2024) and Promptriever (Weller et al., 2025), we first embed instructions, queries, and passages using the off-the-shelf embedding model `E5-Mistral`(Wang et al., 2023a). As shown in Figure 4(a), samples from our `InF-IR` cover a significantly larger semantic space compared to FollowIR and Promptriever. This broader coverage highlights the effectiveness of our synthesized negative instructions and queries in capturing complex semantic variations, crucial for robust contrastive learning.

**Quantitative Analysis of Diversity.** To quantitatively evaluate data diversity, we employ two diversity metrics: *average pairwise sample similarity (APS)* and *inter-sample N-gram frequency*

*(INGF)* (Mishra et al., 2020). Results presented in Figure 4(b) clearly indicate that `InF-IR` achieves superior diversity scores compared to FollowIR and Promptriever, with a lower APS (indicating fewer redundant samples) and a higher INGF (reflecting greater textual diversity).

# 5  `InF-Embed`: INSTRUCTION-AWARE EMBEDDING TRAINING PARADIGM

In this section, we introduce `InF-Embed`, a training framework aimed at improving instruction-aware IR. Specifically, we propose two distinct interactions between instructions and queries (section 5.1), and then further explore various contrastive learning objectives (section 5.2).

## 5.1  INSTRUCTION-QUERY INTERACTION AND REPRESENTATION

We adopt a dual-encoder paradigm (Karpukhin et al., 2020), comprising two encoders $g\left(\cdot\,;\,\theta_P\right)$ and $g\left(\cdot\,;\,\theta_{I,Q}\right)$ to represent corresponding entities within a shared $d$-dimensional embedding space:

$$\mathbf{p}_i = g\left(P_i\,;\,\theta_P\right), \quad \mathbf{i}_i = g\left(I_i\,;\,\theta_{I,Q}\right), \quad \mathbf{q}_i = g\left(Q_i\,;\,\theta_{I,Q}\right), \tag{2}$$

where $\mathbf{p}_i, \mathbf{i}_i, \mathbf{q}_i \in \mathbb{R}^d$ denote the embedding for the passage, instruction, and query, respectively.

**Instruction-Aware Query Representation.** Our primary goal is to improve the instruction-awareness of retrieval models by explicitly incorporating instruction semantics into query representations. To this end, we introduce an instruction-aware query $IQ_{j,k}$ and its embedding $\mathbf{iq}_{j,k}$ designed to integrate instruction-specific context from $I_j$ while interpreting query $Q_k$. We then propose two interaction strategies to compute the combined embedding $\mathbf{iq}$:

$\diamond$ **Interaction I (Self-Attention)**: For each instruction-query pair $(I, Q)$, we concatenate the instruction $I$ with the query $Q$ to construct the instruction-aware query using the simple template of `"<Instruction> <Query>"`. The corresponding embedding $\mathbf{iq}$ is then computed as:

$$\mathbf{iq}_{j,k} = g\left(\texttt{concat}\left(I_j, Q_k\right)\,;\,\theta_{I,Q}\right). \tag{3}$$

When using a decoder-based retriever where $g\left(\cdot\,;\,\theta_{I,Q}\right)$ employs causal attention exclusively, this concatenation naturally allows the model to incorporate instructional context when processing the query. Note that this straightforward approach enables instruction-following retrieval without requiring architectural modifications or introducing additional training parameters.

$\diamond$ **Interaction II (Cross-Attention)**: Although effective, concatenation in Eq. equation 3 can be computationally expensive as it requires a full forward pass for every instruction-query pair. To mitigate this inefficiency, we propose an alternative cross-attention-based mechanism, which explicitly integrates instruction embeddings into the query embeddings via attention:

$$\mathbf{iq}_{j,k} = \texttt{softmax}\left(\left(\mathbf{i}_j \cdot W_{\mathbf{i}}\right)\left(\mathbf{q}_k \cdot W_{\mathbf{q},1}\right)^\top / \sqrt{d}\right)\left(\mathbf{q}_k \cdot W_{\mathbf{q},2}\right), \tag{4}$$

where $W_{\mathbf{i}}, W_{\mathbf{q},1}, W_{\mathbf{q},2} \in \mathbb{R}^{d \times d}$ are learnable linear transformations. We then define the scoring function for retrieval as:

$$s_\theta\left(P_i, I_j, Q_k\right) = \texttt{sim}\left(\mathbf{p}_i, \mathbf{iq}_{j,k}\right), \tag{5}$$

where $\theta = \theta_P \cup \theta_{I+Q}$ denotes parameters from both passage and instruction-query encoders $g\left(\cdot\,;\,\theta_P\right)$ and $g\left(\cdot\,;\,\theta_{I,Q}\right)$; $\texttt{sim}\left(\cdot,\cdot\right)$ represents the cosine similarity between these embeddings.

## 5.2  CONTRASTIVE LEARNING OBJECTIVES

After constructing the positive and negative samples in `InF-IR`, we flatten them into training tuples $(P_i, I_j, Q_k)$, where identical indices $(i = j = k)$ indicate matched positive samples, while differing indices represent unpaired hard negatives. Let the training set be denoted as $\mathcal{D} = \{P_i, I_i, Q_i\}_{i=1}^n$. We then introduce an efficient negative sampling strategy along with two contrastive learning objectives.

**Marginal Sampling Strategy for Negatives.** Direct specializing the general NCE objective (Eq.equation 1) in a multivariate setup involving passages $(P)$, instructions $(I)$, and instruction-aware queries $(IQ)$ results in combinatorial sampling complexity $\mathcal{O}\left(|\mathcal{B}|^{|\mathbf{y}|}\right)$, growing combinatorially for large batch sizes, where $|\mathbf{y}|$ denotes the number of input variables. For instance,

setting $\mathbf{y} = (P, I, IQ)$ yields a cubic summation in the denominator of Eq.equation 1, *i.e.*, $\sum_{m \sim \mathcal{B}} \sum_{j \sim \mathcal{B}} \sum_{k \sim \mathcal{B}} \exp(s_\theta(P_m, I_j, IQ_k))$. To enhance computational efficiency, we propose a marginal negative sampling strategy, independently sampling negatives for each variable in $\mathbf{y}$, while fixing others to their positives. This simplifies the denominator in Eq. equation 1 for a positive example indexed by $i$ as follows:

$$\sum_{m \sim \mathcal{B}} \exp(s_\theta(P_m, I_i, IQ_i)) + \sum_{j \sim \mathcal{B}} \exp(s_\theta(P_i, I_j, IQ_i)) + \sum_{k \sim \mathcal{B}} \exp(s_\theta(P_i, I_i, IQ_k)),$$

reducing complexity from combinatorial to linear, *i.e.*, $\mathcal{O}(|\mathcal{B}| \cdot |\mathbf{y}|)$.

◇ **Objective I (Univariate Conditional Modeling)**: Building upon the conditional probability perspective (section 3), we propose a univariate objective modeling three conditional distributions, $\mathbb{P}(P|I, Q)$, $\mathbb{P}(I|P, Q)$, and $\mathbb{P}(IQ|P)$, via separate contrastive terms:

$$\ell_{P,I,IQ}^{\text{uni}} = -\mathbb{E}_{i \sim \mathcal{B}} \Bigg[ \underbrace{\log \frac{\exp(\text{sim}(\mathbf{p}_i, \mathbf{iq}_{i,i}))}{\sum_{m \sim \mathcal{B}} \exp(\text{sim}(\mathbf{p}_m, \mathbf{iq}_{i,i}))}}_{\ell_P^{\text{uni}} \text{ w.r.t. } \mathbb{P}(P|I,Q)} + \underbrace{\log \frac{\exp(\text{sim}(\mathbf{p}_i, \mathbf{iq}_{i,i}))}{\sum_{j \sim \mathcal{B}} \exp(\text{sim}(\mathbf{p}_i, \mathbf{iq}_{j,i}))}}_{\ell_I^{\text{uni}} \text{ w.r.t. } \mathbb{P}(I|P,Q)} + \underbrace{\log \frac{\exp(\text{sim}(\mathbf{p}_i, \mathbf{iq}_{i,i}))}{\sum_{k \sim \mathcal{B}} \exp(\text{sim}(\mathbf{p}_i, \mathbf{iq}_{k,k}))}}_{\ell_{IQ}^{\text{uni}} \text{ w.r.t. } \mathbb{P}(IQ|P)} \Bigg],$$

(6)

which flexibly enables any combination of univariate conditional modeling by selectively retaining the desired contrastive terms.

◇ **Objective II (Multivariate Conditional Modeling)**: Alternatively, we can ensure instruction-following by keeping instructions as part of the contrasting inputs. This naturally leads to conditional modeling with multivariate inputs such as $(P, I)$, $(P, IQ)$, $(I, IQ)$, and $(P, I, IQ)$ conditioned on the remaining variables. Using the marginal sampling strategy, we formulate the multivariate objective for $(P, I, IQ)$ as:

$$\ell_{P,I,IQ}^{\text{multi}} = -\mathbb{E}_{i \sim \mathcal{B}} \Bigg[ \log \frac{\exp\big(\text{sim}(\mathbf{p}_i, \mathbf{iq}_{i,i})\big)}{\underbrace{\sum_{m \sim \mathcal{B}} \exp\big(\text{sim}(\mathbf{p}_m, \mathbf{iq}_{i,i})\big)}_{\text{Marginal Negatives for } P_i} + \underbrace{\sum_{j \sim \mathcal{B}} \exp\big(\text{sim}(\mathbf{p}_i, \mathbf{iq}_{j,i})\big)}_{\text{Marginal Negatives for } I_i} + \underbrace{\sum_{k \sim \mathcal{B}} \exp\big(\text{sim}(\mathbf{p}_i, \mathbf{iq}_{k,k})\big)}_{\text{Marginal Negatives for } IQ_i}} \Bigg], \quad (7)$$

where other variations of the multivariate objective, such as $\ell_{P,I}^{\text{multi}}$, $\ell_{P,IQ}^{\text{multi}}$, and $\ell_{I,IQ}^{\text{multi}}$, can be readily derived by eliminating the corresponding marginal negatives from the denominator in Eq. equation 7.

Empirically, the univariate contrastive objective in Eq. equation 6 may experience competition among its individual terms. In contrast, the multivariate objective presented in Eq. equation 7 formulates a more challenging ranking-based contrastive task by introducing a larger set of hard negatives that the retriever must effectively differentiate. Consequently, this multivariate formulation potentially exhibits greater robustness to competition-related issues, as evidenced in similar same-tower retrieval contexts (Moiseev et al., 2023; Ren et al., 2021). See additional details in section C.

# 6 EXPERIMENTS

## 6.1 EXPERIMENTS SETUP

**Evaluation Datasets.** We conduct a comprehensive evaluation across the following representative instruction-following retrieval datasets: (1) **FollowIR** (Weller et al., 2024) including *Robust04*, *News21*, and *Core17*, (2) **MAIR** (Sun et al., 2024) including *Dynamic Domain (DD)* and *Fair Ranking (FR)*, and (3) **Bright** (Su et al., 2024). Detailed descriptions are in section D.

**Evaluation Metrics.** Following Weller et al. (2024); Oh et al. (2024), we consider the (1) mean average precision (**MAP**), (2) pairwise mean reciprocal rank (*p*-**MRR**), and (3) normalized discounted cumulative gain (**nDCG@5** for **FollowIR** and **nDCG@10** for **MAIR**) jointly as the metric, while $p$-MRR is used as the main metric to evaluate the effectiveness of the instruction-following retrieval.

**Benchmarks and Baselines.** We compare the following categories of baselines for a comprehensive benchmark evaluation: (1) *non-instruction retrieval models*, (2) *instruction-following retrieval models*, and (3) *instruction-tuned LMs*. We include additional details of baselines in section E.

**Implementation Details.** We consider both embedding models (e5-base-v2, e5-large-v2, ModernBERT-base) and decoder-only LMs (Llamma-3.2 and Qwen-2.5 variants) as backbone LMs for instruction-aware tuning. Additional details of the implementation are available in section F.

Table 2: Main experimental results comparing base models and their variants trained with `InF-Embed` on multiple instruction-following retrieval benchmarks.

| Datasets (→) | Robust04 | | News21 | | Core17 | | FollowIR | | DD-15 | DD-16 | DD-17 | FR-21 | FR-22 | Bright |
|---|---|---|---|---|---|---|---|---|---|---|---|---|---|---|
| Metrics (→) | MAP | p-MRR | nDCG | p-MRR | MAP | p-MRR | score | p-MRR | nDCG | nDCG | nDCG | nDCG | nDCG | nDCG |
| *Base Size: < 1B parameters* | | | | | | | | | | | | | | |
| e5-base-v2 | 13.4 | -6.7 | 20.9 | -2.0 | 14.0 | -2.9 | 16.1 | -3.9 | 40.3 | 31.5 | 32.7 | 29.4 | 61.5 | 3.7 |
| **+InF-Embed** | 14.0 | 6.9 | 23.8 | 3.2 | 11.6 | 5.3 | 16.5 | 5.1 | 47.5 | 35.5 | 32.9 | 49.8 | 78.9 | 8.4 |
| e5-large-v2 | 17.4 | -4.2 | 24.3 | 0.9 | 17.0 | 0.1 | 19.6 | -1.1 | 41.1 | 35.6 | 32.7 | 15.6 | 51.1 | 7.6 |
| **+InF-Embed** | 17.5 | 9.4 | 26.6 | 2.0 | 16.0 | 7.1 | 20.0 | 6.2 | 51.4 | 37.9 | 34.7 | 57.0 | 89.2 | 9.2 |
| ModernBERT-base | 4.29 | -5.8 | 4.3 | -1. | 5.7 | -0.5 | 4.8 | -0.3 | 2.3 | 3.6 | 8.7 | 3.0 | 5.4 | 0.5 |
| **+InF-Embed** | 10.0 | 0.3 | 6.0 | 0.1 | 9.8 | 2.9 | 8.6 | 1.1 | 44.8 | 31.8 | 35.8 | 50.6 | 69.0 | 7.8 |
| *Large Size: 1-5B parameters* | | | | | | | | | | | | | | |
| Llama-3.2-1B | 8.0 | -1.5 | 17.7 | 1.5 | 9.8 | 0.4 | 11.8 | 0.1 | 3.1 | 5.4 | 8.3 | 3.2 | 26.4 | 0.1 |
| **+InF-Embed** | 16.8 | 6.0 | 20.8 | 0.7 | 13.9 | 3.8 | 17.2 | 3.5 | 50.5 | 36.8 | 36.7 | 57.1 | 87.0 | 9.1 |
| Llama-3.2-1B-Inst | 8.6 | -2.1 | 11.1 | 0.6 | 8.7 | 0.2 | 9.5 | -0.4 | 8.3 | 14.9 | 18.3 | 4.6 | 42.1 | 0.4 |
| **+InF-Embed** | 19.1 | 5.6 | 26.1 | 3.8 | 15.2 | 1.9 | 20.2 | 3.8 | 50.7 | 36.5 | 38.9 | 54.6 | 81.4 | 10.9 |
| Qwen2.5-1.5B | 4.7 | -0.5 | 7.5 | -0.2 | 5.9 | 1.6 | 6.0 | 0.3 | 1.0 | 2.7 | 2.4 | 1.5 | 5.5 | 0.2 |
| **+InF-Embed** | 16.8 | 4.9 | 14.1 | 2.7 | 12.7 | 1.9 | 14.5 | 3.2 | 42.0 | 27.2 | 36.0 | 43.5 | 45.2 | 8.5 |
| Qwen2.5-1.5B-Inst | 4.7 | -1.2 | 9.8 | 2.3 | 6.4 | 0.8 | 7.0 | 0.6 | 0.5 | 2.2 | 2.4 | 1.6 | 4.4 | 0.1 |
| **+InF-Embed** | 17.9 | 3.9 | 17.5 | 0.7 | 13.6 | 3.8 | 16.3 | 2.8 | 48.4 | 35.3 | 35.3 | 39.0 | 40.6 | 8.4 |
| Qwen2.5-3B | 5.0 | -0.8 | 8.3 | 0.8 | 5.8 | 1.1 | 6.3 | 0.4 | 1.0 | 3.2 | 2.3 | 1.5 | 7.5 | 0.2 |
| **+InF-Embed** | 17.6 | 4.3 | 19.5 | 1.1 | 12.2 | 3.6 | 16.4 | 3.0 | 49.2 | 30.1 | 34.0 | 53.2 | 75.3 | 10.5 |
| Qwen2.5-3B-Inst | 5.0 | -1.3 | 9.7 | 2.4 | 6.6 | -0.4 | 7.1 | 0.2 | 1.3 | 3.1 | 2.2 | 1.7 | 8.8 | 0.3 |
| **+InF-Embed** | 19.6 | 3.3 | 22.4 | 1.8 | 14.6 | 3.7 | 18.9 | 2.9 | 45.3 | 29.2 | 35.0 | 55.4 | 72.7 | 10.6 |

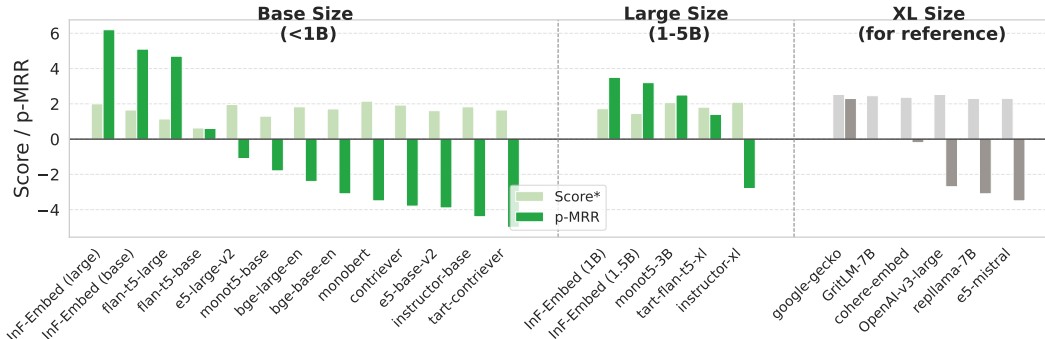

Figure 5: Comparative analysis of instruction-following capabilities on the Follow-IR benchmark across model architectures of varying scales. Models are grouped by parameter count and ranked by p-MRR scores within each category. Standard retrieval metrics (score∗) are normalized by a factor of 10 to facilitate visual comparison with $p$-MRR values.

## 6.2 MAIN EXPERIMENT RESULTS

Table 2 presents comprehensive comparative results between various baselines and their corresponding variants enhanced by our proposed `InF-Embed`. We observe several key findings: (i) Embedding-based models trained on `InF-IR` achieve notable instruction-following improvements ($+1.36@p$-MRR) and enhanced overall retrieval performance; (ii) Auto-regressive LMs, initially limited in retrieval, significantly benefit from `InF-IR`, achieving retrieval effectiveness ($@$nDCG) comparable to similarly sized embedding models; (iii) Fine-tuning previously trained retrievers (*e.g.*, `e5-base-v2`) on `InF-IR` further boosts retrieval scores ($+14.3@$nDCG) and substantially improves instruction-following ability ($+8.2@p$-MRR), highlighting the broad utility and effectiveness of `InF-Embed`.

We also compare our best-performing checkpoints from various backbone models against state-of-the-art retrieval models in Figure 5. The results show that the `InF-Embed` models consistently outperform baseline retrievers of similar size and achieve competitive performance compared to larger-scale or proprietary retrieval models. Additional experimental results are available in section G.

## 6.3 CONFIGURATION AND ABLATION STUDY

**Effect of Objective Function.** Table 3 compares contrastive loss configurations (section 5.2) on the FollowIR benchmark, yielding four main insights: (i) *Multivariate contrastive loss ($\ell_{P,I}^{multi}$)* outperforms other variants, underscoring the importance of simultaneously contrasting instructions and passages as introduced in `InF-IR`. Instruction contrast explicitly guides instruction understanding,

Table 3: Configuration comparison on Follow-IR (Weller et al., 2024) benchmarking multiple loss function designs and varying sizes of backbone LMs.

| Category (→) | Encoder | | Decoder | | | | | | | | | | | |
| --- | --- | --- | --- | --- | --- | --- | --- | --- | --- | --- | --- | --- | --- | --- |
| Base Model (→) Model Size (→) | ModernBERT 109M | | Llama-3.2 1B | | Llama-3.2 1B-Instruct | | Qwen2.5 1.5B | | Qwen2.5 1.5B-Instruct | | Qwen2.5 3B | | Qwen2.5 3B-Instruct | |
| Config. (↓) | score | p-MRR | score | p-MRR | score | p-MRR | score | p-MRR | score | p-MRR | score | p-MRR | score | p-MRR |
| Base | 4.76 | -0.27 | 11.84 | 0.13 | 9.45 | -0.43 | 6.03 | 0.29 | 6.98 | 0.63 | 6.33 | 0.38 | 7.07 | 0.24 |
| w/ $\ell_P^{uni}$ | 9.70 | -0.22 | 17.15 | **3.48** | 19.81 | 3.76 | 14.28 | 2.27 | **16.34** | **2.76** | **16.46** | 1.12 | 16.61 | 2.46 |
| w/ $\ell_I^{uni}$ | 3.58 | -1.00 | 9.17 | -0.48 | 10.39 | -0.10 | 9.29 | -0.18 | 8.34 | -0.73 | 7.91 | -1.76 | 8.00 | -1.01 |
| w/ $\ell_{IQ}^{uni}$ | 9.39 | 0.02 | 18.69 | 1.82 | 17.59 | 2.98 | 12.94 | 1.69 | 14.03 | 1.98 | 14.62 | 0.71 | 17.45 | 0.86 |
| w/ $\ell_{P,I}^{uni}$ | 8.25 | 1.08 | 17.98 | 2.11 | 19.59 | 2.89 | 14.38 | 2.00 | 15.12 | 1.61 | 15.34 | 2.87 | 18.40 | 1.27 |
| w/ $\ell_{P,IQ}^{uni}$ | 9.30 | -0.03 | 17.23 | 1.12 | 18.48 | 3.21 | 13.64 | 1.46 | 14.05 | 2.62 | 15.57 | 2.08 | 16.73 | 2.13 |
| w/ $\ell_{I,IQ}^{uni}$ | 7.51 | 0.43 | 19.12 | 1.36 | 19.00 | 1.50 | 13.14 | 1.24 | 13.27 | 1.66 | 15.03 | 0.87 | 15.61 | -0.08 |
| w/ $\ell_{P,I,IQ}^{uni}$ | 8.61 | 0.84 | 17.96 | 1.42 | 19.98 | 2.58 | 13.83 | 1.49 | 14.11 | 2.64 | 16.14 | 2.68 | 17.30 | 1.26 |
| w/ $\ell_{P,I}^{multi}$ | **13.27** | 0.09 | 19.05 | 2.30 | **20.15** | 3.76 | **14.52** | **3.20** | 15.18 | 2.51 | 16.41 | **3.00** | **18.87** | **2.94** |
| w/ $\ell_{P,IQ}^{multi}$ | 9.05 | 0.30 | 17.57 | 2.00 | 17.71 | 3.49 | 13.18 | 2.07 | 14.06 | 2.65 | 15.49 | 1.49 | 17.50 | 2.32 |
| w/ $\ell_{I,IQ}^{multi}$ | 8.36 | -0.12 | **19.21** | 0.38 | 19.22 | 1.63 | 14.07 | 1.09 | 13.61 | 2.20 | 14.00 | 1.32 | 16.03 | 1.20 |
| w/ $\ell_{P,I,IQ}^{multi}$ | 8.58 | **1.09** | 18.75 | 2.11 | 19.76 | 2.31 | 12.83 | 1.65 | 14.22 | 2.65 | 16.25 | 1.44 | 17.62 | 2.25 |
| Attn Base | 3.66 | -0.25 | 5.53 | 0.83 | 6.68 | 0.59 | 4.07 | 0.23 | 4.30 | 0.02 | 3.89 | 0.21 | 3.98 | -0.03 |
| Attn Best | 8.57 | 0.42 | 9.14 | 1.66 | 11.30 | 0.74 | 13.80 | 1.10 | 14.05 | 2.13 | 15.59 | 0.68 | 11.29 | 0.76 |

while passage contrast strengthens alignment between instruction-conditioned queries and relevant passages; (ii) *Multivariate objectives consistently surpass simpler univariate objectives*, highlighting the necessity of jointly modeling interactions among instructions, queries, and passages to improve instruction-aware retrieval; (iii) *Decoder-only models outperform encoder-only models* in both retrieval quality and instruction-following, likely due to larger parameter capacity and richer pretraining data, enabling better handling of complex instruction-based scenarios; and (iv) *Joint encoding of instruction and query (concatenation) surpasses separate encoding (attention-based)*, benefiting from the autoregressive modeling capabilities of decoder-only architectures. However, joint encoding complicates partial contrastive training. Thus, separately encoding instructions, queries, and passages via attention may offer a more flexible and efficient approach for future contrastive objective designs.

Table 4: Different designs in `InF-Embed`.

| Config. (→) | Share Encoder | Pooling | Epoch | p-MRR |
| --- | --- | --- | --- | --- |
| Qwen2.5-1.5B | ✓ | last | **2** | **2.27** |
| Qwen2.5-1.5B | ✓ | avg. | 2 | -0.39 |
| Qwen2.5-1.5B | ✗ | last | 2 | 0.26 |
| Qwen2.5-1.5B | ✓ | last | 1 | -0.06 |

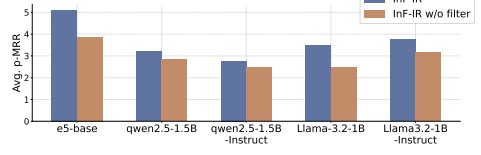

Figure 6: Effect of quality filtering.

**Effect of Negative Pairs Synthesis.** We analyze the impact of various training configurations in Table 4, using `Qwen2.5-1.5B` as the base model. For decoder-only LLMs, our results indicate that using a shared encoder for instruction-aware queries and passages, combined with last-token pooling, consistently yields the best performance and is thus recommended as the default configuration.

**Effect of Quality Check.** We investigate the effectiveness of our data quality-check step by comparing model performance trained on the original unfiltered data versus our quality-filtered `InF-IR` (Figure 6). Despite the unfiltered dataset being substantially larger, its lower data quality significantly degrades model performance under identical training conditions. This highlights the critical importance of rigorous data validation in our synthesis pipeline.

# 7 CONCLUSION

In this paper, we introduce `InF-IR`, a large-scale, high-quality training corpus explicitly designed to enhance instruction-following retrieval models. Built upon `InF-IR`, `InF-Embed` demonstrates robust improvement in instruction-following capabilities across multiple retrieval benchmarks, achieving substantial performance gains for both embedding-based (+9.0 p-MRR) and auto-regressive language models (+4.2 p-MRR). In addition, we provide a systematic benchmarking of contrastive learning objectives across various model architectures and sizes, establishing best practices that will accelerate the development of next-generation instruction-following retrieval systems. An important line of future work is to extend `InF-IR` and `InF-Embed` to reasoning-intensive retrieval models (Chen et al., 2025; Jin et al., 2025; Guan et al., 2025) in instruction-following IR.

## ETHICS STATEMENT

All authors have read and followed the Code of Ethics. Our work uses public corpora (MS MARCO and TREC) and synthetic text produced by LLMs. We respect the licenses of the source datasets. We do not collect or release any personally identifiable information. When using hosted LLM APIs to synthesize instructions and negatives, we followed the provider's data-use policies and opted out of human review according to the Azure OpenAI Additional Use Case Form. A small human annotation study approved by IRB was conducted to check agreement with model judgments. Annotators were three computer science major students; they received task instructions and examples, and they worked only with public text passages and synthetic prompts. No demographic or sensitive attributes were collected. The study did not involve medical, financial, or other sensitive content.

We are aware of risks related to bias and potential misuse. Synthetic data may reflect biases present in web-scale models, and stronger instruction-following retrieval could be misused to surface harmful content. To reduce these risks, we (i) filtered generations that drift from the intended task or include unsafe content, (ii) validated positives and hard negatives with an independent reasoning model to favor clear, task-relevant pairs, and (iii) will release data and checkpoints under a research license that prohibits misuse and prohibits attempts to target individuals or protected classes. We also checked for test-set contamination by string matching between our training data and the evaluation sets and did not observe overlaps.

## REPRODUCIBILITY STATEMENT

We designed the paper, appendix, and supplement to support full replication. Data curation steps, including instruction synthesis, query poisoning, and hard-negative construction, are specified in section 4.1. The prompts used to generate instructions and queries are provided verbatim in Appendix H. Rule-based filters and the model-based quality check are described in section 4.1 and Appendix D, and the agreement study setup appears in Appendix F. Model architectures, interaction mechanisms, and training losses are given in section 6 with complete loss definitions in Appendix B. We list datasets, splits, and metrics in section 6 and Appendix F. Hyperparameters, optimizer settings, batching, pooling choices, hardware, and training durations are documented in Appendix F.

The supplementary materials include an anonymous code repository with: scripts to recreate the synthetic triples from the licensed sources, the exact prompts, configuration files for each backbone, seeds, and evaluation code. Because MS MARCO and some TREC sources cannot be redistributed, we provide document identifiers and instructions that download the originals from their hosts, followed by our preprocessing scripts. We also include instructions to run the two instruction–query interaction variants and both contrastive objectives. Pretrained checkpoints will be shared for research use under terms consistent with the upstream licenses.

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

# Appendix

## A LIMITATIONS AND BROADER IMPACTS

### A.1 LIMITATIONS

While our proposed dataset and methodology substantially advance instruction-following capabilities in retrieval models, several limitations must be acknowledged: First, our negative sample generation and rigorous quality checks involve advanced reasoning models like o3-mini, which are computationally intensive. Researchers with limited computational resources might find reproducing or extending our dataset challenging. Secondly, our dataset primarily extends MS MARCO, which is a general-domain dataset. While we demonstrate improvements across multiple general-domain benchmarks, the effectiveness of our approach in highly specialized or domain-specific retrieval tasks may require additional investigation and potential adaptation. Thirdly, although our marginal sampling strategy significantly reduces complexity, scaling the multivariate contrastive objectives to extremely large batch sizes or significantly larger model scales remains nontrivial. Future research could explore further optimization techniques to enhance scalability.

### A.2 BROADER IMPACTS

**Potential Positive Societal Impacts.** Our contributions significantly improve the capability of IR systems to accurately interpret and follow user instructions. This enhancement can lead to higher efficiency and precision in information retrieval tasks across various real-world applications, including personalized web search, educational content discovery, and knowledge-intensive professional settings. By ensuring retrieval outputs closely align with explicit user instructions, InF-IR and InF-Embed contribute to reducing user effort and frustration, thereby positively influencing user experience and productivity.

**Potential Negative Societal Impacts.** Improved instruction-following retrieval systems could inadvertently amplify existing biases or misinformation if the training data inherently contains biased or incorrect information. Given that InF-IR and InF-Embed leverage synthetic generation techniques and LLMs trained on web-scale data, there remains a risk of propagating undesirable stereotypes or inaccuracies present in these sources. Additionally, enhanced retrieval models might facilitate misuse, such as targeted misinformation dissemination or unauthorized data retrieval, emphasizing the need for responsible deployment and continual oversight.

### A.3 DATA PRIVACY AND LICENSING

InF-IR creation relies on publicly available datasets, specifically MS MARCO and datasets from the TREC collections, each of which comes with specific licensing terms that we have strictly followed. MS MARCO is distributed under a non-commercial license, and any derived datasets, including ours, must adhere to similar terms. Researchers aiming to use InF-IR and InF-Embed should ensure compliance with the respective licenses of these original sources. Additionally, while leveraging LLMs such as gpt-4o-mini and o3-mini to generate synthetic instructions and queries, we have carefully avoided generating personally identifiable or sensitive information. Nevertheless, practitioners must exercise caution when extending our methods to datasets involving sensitive or private data, ensuring strict adherence to data privacy regulations and ethical standards relevant to their application contexts.

### A.4 ETHICAL STATEMENTS

Throughout our dataset creation and model training, we strictly adhered to the licensing agreements of all source datasets (*e.g.*, MS MARCO and TREC collections). In addition, we conducted thorough checks to prevent any form of contamination of the test set. Despite these measures, practitioners using our dataset and methods must ensure compliance with privacy standards and ethical guidelines relevant to their specific applications, especially when dealing with sensitive or user-specific data. InF-IR involves the usage of OpenAI APIs. To prevent any potential information leakage, we strictly follow data usage guidelines of Microsoft Azure's Open AI API service and have withdrawn from the human review process by completing and submitting the Azure OpenAI Additional Use Case Form. We do not foresee other ethics issues.

## B  InF-IR: ADDITIONAL DATA CURATION DETAILS

### B.1  ADDITIONAL DATA QUALITY CHECKS

Beyond the quality assurance steps described in Section 4.1, we employ a rule-based filtering step to further enhance sample quality. Specifically, we truncate synthetic instructions and queries at the first occurrence of a newline character (\n), ensuring the removal of irrelevant or extraneous text.

### B.2  ADDITIONAL DATA SOURCES

While the primary InF-IR builds upon MS MARCO, we are also extending our curation approach to additional datasets, including TREC Robust 2004 (Voorhees, 2004), Leetcode, and MetaMath (Yu et al., 2023). For Robust 2004, we utilize the identical data curation process applied to MS MARCO. In the case of Leetcode and MetaMath, queries are derived from problem descriptions, while passages correspond to their respective solutions. Synthetic instructions for Leetcode explicitly include the programming language and specify the intended technical solution approach. For MetaMath problems, instructions emphasize the mathematical strategy or method. When generating negative examples for Leetcode, we ensure the programming language (e.g., Python, Java, C++) remains constant between positive and negative instructions to prevent the model from relying solely on language cues to differentiate samples. All other data curation procedures remain consistent across datasets.

## C  InF-Embed: ADDITIONAL METHOD DETAILS

### C.1  UNIVARIATE CONTRASTIVE LOSS DETAILS

Here are the detailed definitions of univariate contrastive objective functions:

$$\ell_P^{\text{uni}} = -\mathbb{E}_{i \sim \mathcal{B}} \left[ \log \frac{\exp(\text{sim}(\mathbf{p}_i, \mathbf{iq}_{i,i}))}{\sum_{m \sim \mathcal{B}} \exp(\text{sim}(\mathbf{p}_m, \mathbf{iq}_{i,i}))} \right], \tag{8}$$

$$\ell_I^{\text{uni}} = -\mathbb{E}_{i \sim \mathcal{B}} \left[ \log \frac{\exp(\text{sim}(\mathbf{p}_i, \mathbf{iq}_{i,i}))}{\sum_{j \sim \mathcal{B}} \exp(\text{sim}(\mathbf{p}_i, \mathbf{iq}_{j,i}))} \right], \tag{9}$$

$$\ell_{IQ}^{\text{uni}} = -\mathbb{E}_{i \sim \mathcal{B}} \left[ \log \frac{\exp(\text{sim}(\mathbf{p}_i, \mathbf{iq}_{i,i}))}{\sum_{k \sim \mathcal{B}} \exp(\text{sim}(\mathbf{p}_i, \mathbf{iq}_{k,k}))} \right], \tag{10}$$

$$\ell_{P,I}^{\text{uni}} = -\mathbb{E}_{i \sim \mathcal{B}} \left[ \log \frac{\exp(\text{sim}(\mathbf{p}_i, \mathbf{iq}_{i,i}))}{\sum_{m \sim \mathcal{B}} \exp(\text{sim}(\mathbf{p}_m, \mathbf{iq}_{i,i}))} + \log \frac{\exp(\text{sim}(\mathbf{p}_i, \mathbf{iq}_{i,i}))}{\sum_{j \sim \mathcal{B}} \exp(\text{sim}(\mathbf{p}_i, \mathbf{iq}_{j,i}))} \right], \tag{11}$$

$$\ell_{P,IQ}^{\text{uni}} = -\mathbb{E}_{i \sim \mathcal{B}} \left[ \log \frac{\exp(\text{sim}(\mathbf{p}_i, \mathbf{iq}_{i,i}))}{\sum_{m \sim \mathcal{B}} \exp(\text{sim}(\mathbf{p}_m, \mathbf{iq}_{i,i}))} + \log \frac{\exp(\text{sim}(\mathbf{p}_i, \mathbf{iq}_{i,i}))}{\sum_{k \sim \mathcal{B}} \exp(\text{sim}(\mathbf{p}_i, \mathbf{iq}_{k,k}))} \right], \tag{12}$$

$$\ell_{P,I,IQ}^{\text{uni}} = -\mathbb{E}_{i \sim \mathcal{B}} \left[ \log \frac{\exp(\text{sim}(\mathbf{p}_i, \mathbf{iq}_{i,i}))}{\sum_{j \sim \mathcal{B}} \exp(\text{sim}(\mathbf{p}_i, \mathbf{iq}_{j,i}))} + \log \frac{\exp(\text{sim}(\mathbf{p}_i, \mathbf{iq}_{i,i}))}{\sum_{k \sim \mathcal{B}} \exp(\text{sim}(\mathbf{p}_i, \mathbf{iq}_{k,k}))} \right], \tag{13}$$

$$\ell_{P,I,IQ}^{\text{uni}} = -\mathbb{E}_{i \sim \mathcal{B}} \left[ \log \frac{\exp(\text{sim}(\mathbf{p}_i, \mathbf{iq}_{i,i}))}{\sum_{m \sim \mathcal{B}} \exp(\text{sim}(\mathbf{p}_m, \mathbf{iq}_{i,i}))} + \log \frac{\exp(\text{sim}(\mathbf{p}_i, \mathbf{iq}_{i,i}))}{\sum_{j \sim \mathcal{B}} \exp(\text{sim}(\mathbf{p}_i, \mathbf{iq}_{j,i}))} + \log \frac{\exp(\text{sim}(\mathbf{p}_i, \mathbf{iq}_{i,i}))}{\sum_{k \sim \mathcal{B}} \exp(\text{sim}(\mathbf{p}_i, \mathbf{iq}_{k,k}))} \right], \tag{14}$$

### C.2  MULTIVARIATE CONTRASTIVE LOSS DETAILS

Here are the detailed definitions of multivariate contrastive objective functions:

$$\ell_P^{\text{multi}} = -\mathbb{E}_{i \sim \mathcal{B}} \left[ \log \frac{\exp\big(\text{sim}(\mathbf{p}_i, \mathbf{iq}_{i,i})\big)}{\sum_{m \sim \mathcal{B}} \exp\big(\text{sim}(\mathbf{p}_m, \mathbf{iq}_{i,i})\big)} \right], \tag{15}$$

$$\ell_I^{\text{multi}} = -\mathbb{E}_{i \sim \mathcal{B}} \left[ \log \frac{\exp\big(\text{sim}(\mathbf{p}_i, \mathbf{iq}_{i,i})\big)}{\sum\limits_{j \sim \mathcal{B}} \exp\big(\text{sim}(\mathbf{p}_i, \mathbf{iq}_{j,i})\big)} \right], \tag{16}$$

$$\ell_{IQ}^{\text{multi}} = -\mathbb{E}_{i \sim \mathcal{B}} \left[ \log \frac{\exp\big(\text{sim}(\mathbf{p}_i, \mathbf{iq}_{i,i})\big)}{\sum\limits_{k \sim \mathcal{B}} \exp\big(\text{sim}(\mathbf{p}_i, \mathbf{iq}_{k,k})\big)} \right], \tag{17}$$

$$\ell_{P,I}^{\text{multi}} = -\mathbb{E}_{i \sim \mathcal{B}} \left[ \log \frac{\exp\big(\text{sim}(\mathbf{p}_i, \mathbf{iq}_{i,i})\big)}{\sum\limits_{m \sim \mathcal{B}} \exp\big(\text{sim}(\mathbf{p}_m, \mathbf{iq}_{i,i})\big) + \sum\limits_{j \sim \mathcal{B}} \exp\big(\text{sim}(\mathbf{p}_i, \mathbf{iq}_{j,i})\big)} \right], \tag{18}$$

$$\ell_{P,IQ}^{\text{multi}} = -\mathbb{E}_{i \sim \mathcal{B}} \left[ \log \frac{\exp\big(\text{sim}(\mathbf{p}_i, \mathbf{iq}_{i,i})\big)}{\sum\limits_{m \sim \mathcal{B}} \exp\big(\text{sim}(\mathbf{p}_m, \mathbf{iq}_{i,i})\big) + \sum\limits_{k \sim \mathcal{B}} \exp\big(\text{sim}(\mathbf{p}_i, \mathbf{iq}_{k,k})\big)} \right], \tag{19}$$

$$\ell_{I,IQ}^{\text{multi}} = -\mathbb{E}_{i \sim \mathcal{B}} \left[ \log \frac{\exp\big(\text{sim}(\mathbf{p}_i, \mathbf{iq}_{i,i})\big)}{\sum\limits_{j \sim \mathcal{B}} \exp\big(\text{sim}(\mathbf{p}_i, \mathbf{iq}_{j,i})\big) + \sum\limits_{k \sim \mathcal{B}} \exp\big(\text{sim}(\mathbf{p}_i, \mathbf{iq}_{k,k})\big)} \right], \tag{20}$$

$$\ell_{P,I,IQ}^{\text{multi}} = -\mathbb{E}_{i \sim \mathcal{B}} \left[ \log \frac{\exp\big(\text{sim}(\mathbf{p}_i, \mathbf{iq}_{i,i})\big)}{\sum\limits_{m \sim \mathcal{B}} \exp\big(\text{sim}(\mathbf{p}_m, \mathbf{iq}_{i,i})\big) + \sum\limits_{j \sim \mathcal{B}} \exp\big(\text{sim}(\mathbf{p}_i, \mathbf{iq}_{j,i})\big) + \sum\limits_{k \sim \mathcal{B}} \exp\big(\text{sim}(\mathbf{p}_i, \mathbf{iq}_{k,k})\big)} \right], \tag{21}$$

## D  EVALUATION DATASET DETAILS

Here are the details for each instruction-following retrieval dataset used in our experiments:

- **FollowIR** (Weller et al., 2024) assesses IR models based on their responsiveness to detailed and realistic instructions extracted from TREC narrative annotations. These narratives encompass explicit inclusion and exclusion criteria. It features queries sourced from TREC Robust 2004 (Voorhees, 2004), TREC Common Core 2017 (Allan et al., 2017), and TREC News 2021 (Soboroff, 2022), enriched with professional narrative annotations and further refined through targeted human reviews. It employs pairwise annotations to effectively measure models' adaptability to evolving instructions.

- **MAIR** (Sun et al., 2024) provides a comprehensive evaluation of instruction-tuned IR models across 126 distinct tasks spanning multiple domains such as academic literature, code retrieval, legal documents, finance, and medical search. It incorporates 10,038 queries paired with 805 distinct instructions sourced from public datasets, TREC tracks, and established IR benchmarks. Each task features meticulous manual annotations that define relevance across diverse query-document-instruction contexts.

- **Bright** (Su et al., 2024) presents reasoning-intensive retrieval tasks that extend beyond conventional lexical or semantic matching, incorporating complex scenarios from diverse domains such as coding, mathematics, economics, and science. Bright contains 1,384 real-world queries drawn from 12 varied datasets, including StackExchange, LeetCode, and AoPS, among others. Documents consist of referenced web pages, programming syntax manuals, and solution explanations unified by shared logical, algorithmic, or theoretical foundations. Relevance labels are human-validated, ensuring alignment with intricate reasoning criteria.

## E  BASELINE DETAILS

Here are the details for each instruction-following retrieval baseline used in our experiments:

- **Contriever** (Izacard et al., 2021) is a bi-encoder dense retriever trained via unsupervised contrastive learning on large text corpora, providing general-purpose semantic representations for zero-shot retrieval.

- **FLAN-T5** (Chung et al., 2022) is an instruction-finetuned variant of T5 designed for zero-shot generalization. It employs an encoder-decoder architecture to generate relevance judgments through prompting without retrieval-specific fine-tuning.

- **E5** (Wang et al., 2022a) models (base and large) are dual-encoder retrieval systems fine-tuned on extensive weakly supervised contrastive pairs. They excel in embedding-based retrieval tasks, explicitly leveraging query and passage instructions for generalized semantic representation.

- **MonoT5** (Nogueira et al., 2020) is a cross-encoder reranker using the T5 framework to jointly model queries and documents, producing highly accurate relevance scores through generation-based prompting.

- **Bge** (Xiao et al., 2024) employs RoBERTa-based dual encoders fine-tuned with contrastive learning, optimized for stable and accurate dense retrieval without explicit task instructions.

- **Instructor** (Oh et al., 2024) generates task-specific embeddings conditioned on natural language instructions, trained with contrastive learning across diverse NLP tasks, allowing flexible zero-shot application in retrieval and similarity tasks.

- **Tart-contriever** (Asai et al., 2022) extends Contriever with instruction-aware embedding generation via multi-task distillation, enhancing zero-shot retrieval capabilities across varied domains.

- **GritLM** (Muennighoff et al., 2024) integrates generative and embedding-based tasks into a single LLaMA-based instruction-tuned model, achieving state-of-the-art embedding benchmarks while supporting flexible instruction-based retrieval.

- **Repllama** (Ma et al., 2024) fine-tunes the LLaMA-2 model for dense retrieval, leveraging contrastive training on retrieval tasks to encode comprehensive document-level information into embeddings, demonstrating strong zero-shot retrieval performance.

## F IMPLEMENTATION DETAILS

### F.1 ADDITIONAL IMPLEMENTATION DETAILS

Model training and testing are conducted on 8 NVIDIA A100 80G GPUs. We use the AdamW optimizer with an initial learning rate of $5 \times 10^{-5}$ for both embedding models and LMs. The batch size is set to 4 per device. To prevent test set contamination (Oren et al., 2023) in external evaluations, we have conducted a string-matching analysis, where we *do not observe any overlap* between the training data in `InF-IR` and the evaluation datasets utilized in this study.

### F.2 HUMAN AGREEMENT STUDY DETAILS

To rigorously validate the reliability and effectiveness of our data quality-check procedure, we performed a human annotation study involving expert annotators. We invite 3 collaborators and coauthors to attend the annotation, including 1 senior graduate student, a junior graduate student, and 1 undergraduate student. All three students CS majored and are all familiar with information retrieval tasks to independently assess a subset of our dataset. Annotators were presented with a randomly sampled selection of instruction-query pairs paired with passages including the original positive passages, synthetically generated negative passages, and randomly sampled in-batch negative distractors from MS MARCO. Each annotator independently identified the passage they thought most relevant to the given instruction-query context. To ensure high annotation quality, all annotator underwent training sessions involving clear task instructions and illustrative examples prior to beginning the main annotation task. Then, we feed the same data to the LLM-based annotators, including `o3-mini` used in this work and other design choices of `gpt-4o` and `gpt-4o-mini`.

We computed pairwise agreement scores between human annotators and LLMs using Cohen's Kappa statistics, which measures inter-rater reliability while accounting for agreement occuring by chance. Subsequently, we computed the average consistency between human judgments and predictions from several large language models, including `o3-mini`, `FollowIR-7B`, `gpt-4o-mini`, and `gpt-4o`. As shown in Figure 3, `o3-mini` consistently achieved the highest Cohen's Kappa scores with human annotators, outperforming the other models evaluated. These results underline the alignment of `o3-mini`'s judgments with human intuition and confirm the robustness and effectiveness of our automated filtering strategy for maintaining high-quality synthetic datasets.

# G   ADDITIONAL EXPERIMENTAL RESULTS

## G.1   ADDITIONAL INSTRUCTION-FOLLOWING IR RESULTS

Table 5: Additional results of various baselines on multiple instruction-following IR datasets.

| Evaluation Datasets (→) | Robust04 | | News21 | | Core17 | | FollowIR | | DD-15 | DD-16 | DD-17 | FR-21 | FR-22 | MAIR |
| Baselines (↓) / Metrics (→) | MAP | p-MRR | nDCG | p-MRR | MAP | p-MRR | score | p-MRR | nDCG | nDCG | nDCG | nDCG | nDCG | nDCG |
|---|---|---|---|---|---|---|---|---|---|---|---|---|---|---|
| *Sparse Retrieval* | | | | | | | | | | | | | | |
| BM25 (2009) | 12.1 | -3.1 | 19.3 | -2.1 | 8.1 | -1.1 | 13.2 | -2.1 | – | – | – | – | – | – |
| *Base Size: < 1B parameters* | | | | | | | | | | | | | | |
| e5-base-v2 (109M) (2022a) | 13.4 | -6.7 | 20.9 | -2.0 | 14.0 | -2.9 | 16.1 | -3.9 | 40.3 | 31.5 | 32.7 | 29.4 | 61.5 | 39.1 |
| **InF-Embed (e5-base-v2)** | 14.0 | 6.9 | 23.8 | 3.2 | 11.6 | 5.3 | 16.5 | 5.1 | 47.5 | 35.5 | 32.9 | 49.8 | 78.9 | 48.9 |
| contriever (109M) (2021) | 19.7 | -6.1 | 22.9 | -2.8 | 15.3 | -2.5 | 19.3 | -3.8 | – | – | – | – | – | – |
| bge-base-en(v1.0/1.5) (109M) (2024) | 16.8 | -6.5 | 20.0 | -0.1 | 14.6 | -2.7 | 17.1 | -3.1 | 21.0 | 16.7 | 33.5 | 25.1 | 29.6 | 25.2 |
| tart-contriever (109M) (2022) | 14.3 | -9.0 | 21.8 | -3.0 | 13.3 | -3.0 | 16.5 | -5.0 | – | – | – | – | – | – |
| instructor-base (109M) (2023) | 17.2 | -10.4 | 22.1 | -1.8 | 15.5 | -1.1 | 18.3 | -4.4 | – | – | – | – | – | – |
| monot5-base (220M) (2020) | 15.7 | -6.2 | 11.0 | 5.0 | 12.2 | -4.1 | 13.0 | -1.8 | 46.7 | 28.5 | 31.8 | 18.3 | 68.5 | – |
| flan-t5-base (248M) (2022) | 6.4 | 5.3 | 6.1 | -0.1 | 6.5 | -3.3 | 6.3 | 0.6 | – | – | – | – | – | – |
| monobert (330M) (2020) | 21.0 | -9.4 | 25.1 | -0.8 | 18.4 | -0.2 | 21.5 | -3.5 | – | – | – | – | – | – |
| e5-large-v2 (330M) (2022a) | 17.4 | -4.2 | 24.3 | 0.9 | 17.0 | 0.1 | 19.6 | -1.1 | 41.1 | 35.6 | 32.7 | 15.6 | 51.1 | 35.2 |
| **InF-Embed (e5-large-v2)** | 17.49 | 9.4 | 26.6 | 2.0 | 16.0 | 7.1 | 20.0 | 6.2 | 51.4 | 37.9 | 34.7 | 57.0 | 89.2 | 54.0 |
| bge-large-en (335M) (2024) | 17.5 | -7.8 | 22.3 | 0.6 | 15.0 | 0.1 | 18.3 | -2.4 | 18.8 | 22.9 | 35.5 | 17.8 | 26.3 | 24.3 |
| flan-t5-large (783M) (2022) | 14.7 | 3.9 | 8.0 | 8.9 | 11.4 | 1.3 | 11.4 | 4.7 | – | – | – | – | – | – |
| *Large Size: 1-5B parameters* | | | | | | | | | | | | | | |
| **InF-Embed (Llama-3.2-1B)** | 16.8 | 6.0 | 20.8 | 0.7 | 13.9 | 3.8 | 17.2 | 3.5 | 50.5 | 36.8 | 36.7 | 57.1 | 87.0 | 53.6 |
| **InF-Embed (Qwen2.5-1.5B)** | 16.8 | 4.9 | 14.1 | 2.7 | 12.7 | 1.9 | 14.5 | 3.2 | 44.2 | 23.4 | 35.4 | 52.6 | 85.1 | 48.1 |
| instructor-xl (1.5B) (2023) | 19.7 | -8.1 | 26.1 | -0.9 | 16.8 | 0.7 | 20.9 | -2.8 | – | – | – | – | – | – |
| tart-flan-t5-xl (2.85B) (2022) | 24.6 | -0.7 | 12.8 | 2.0 | 17.0 | 2.8 | 18.1 | 1.4 | – | – | – | – | – | – |
| monot5-3B (2020) | 27.3 | 4.0 | 16.5 | 1.8 | 18.2 | 1.8 | 20.7 | 2.5 | – | – | – | – | – | – |
| *XL Size and Proprietary LLMs: >5B parameters (for reference)* | | | | | | | | | | | | | | |
| e5-mistral (7B) (2023a) | 23.1 | -9.6 | 27.8 | -0.9 | 18.3 | 0.1 | 23.1 | -3.5 | 50.3 | 33.7 | 35.1 | 58.3 | 84.8 | 52.4 |
| **InF-Embed (e5-mistral)** | 25.5 | 6.2 | 23.9 | 1.5 | 23.0 | 6.3 | 24.1 | 4.7 | 52.0 | 37.3 | 37.4 | 58.4 | 89.1 | 54.8 |
| Qwen2.5-7B | 10.1 | 1.0 | 13.8 | 3.1 | 7.3 | -0.3 | 10.4 | 1.3 | 2.6 | 5.5 | 3.4 | 1.9 | 12.9 | 5.3 |
| **InF-Embed (Qwen2.5-7B)** | 26.7 | 6.4 | 25.6 | 1.8 | 23.4 | 6.5 | 25.2 | 4.9 | 47.6 | 32.1 | 36.8 | 51.5 | 86.6 | 50.9 |
| GritLM-7B (2024) | 28.6 | -1.7 | 24.4 | -1.0 | 20.8 | 2.6 | 24.6 | -0.0 | 52.3 | 36.0 | 36.3 | 58.3 | 82.7 | 53.1 |
| NV-Embed-v1 (7B) (2024) | – | – | – | – | – | – | – | – | 45.0 | 31.5 | 30.8 | 43.0 | 84.7 | 47.0 |
| repllama-v1-7b (2024) | 24.0 | -8.9 | 24.5 | -1.8 | 20.6 | 1.3 | 23.0 | -3.1 | – | – | – | – | – | – |
| promptriever-llama2-7b (2025) | 28.3 | 11.7 | 28.5 | 6.4 | 21.6 | 15.4 | 26.1 | 11.2 | – | – | – | – | – | – |
| OpenAI-v3-large | 27.2 | -5.8 | 27.2 | -2.0 | 21.6 | -0.2 | 25.3 | -2.7 | – | – | – | – | – | – |
| cohere-embed-english-v3.0 | 22.3 | -3.6 | 28.3 | 0.2 | 20.6 | 2.8 | 23.7 | -0.2 | – | – | – | – | – | – |
| google-gecko (2024) | 23.3 | -2.4 | 29.5 | 3.9 | 23.2 | 5.4 | 25.3 | 2.3 | – | – | – | – | – | – |

Table 5 presents additional results of various baselines on instruction-following IR datasets. Key additional insights from this evaluation include:

• **Sparse vs. Dense Retrieval.** Dense retrieval models consistently outperform traditional sparse retrieval methods such as BM25, particularly in instruction-following tasks, highlighting the advantage of semantic embedding-based approaches.

• **Model Size and Effectiveness.** Larger model sizes generally exhibit stronger retrieval and instruction-following performance. Models in the XL size category (over 5B parameters), such as GritLM-7B and promptriever-llama2-7b, deliver state-of-the-art results, demonstrating the benefit of increased parameter count and training scale.

• **Impact of Instruction-Tuning.** Instruction-tuned model*e.g.*, FLAN-T5, Instructor, and GritLM) significantly outperform models without explicit instruction-tuning. These improvements underscore the critical role of task-specific instructions in enhancing retrieval capabilities and aligning model outputs more closely with user intentions.

## G.2   ADDITIONAL CONFIGURATION BENCHMARKS

Table 6 benchmarks various contrastive loss configurations (Section 5.2) with detailed comparisons on the FollowIR dataset. Our key observations are as follows:

• **Contrastive Loss.** Models trained with $\ell_{P,I}^{\text{multi}}$ achieve the highest performance. This result highlights the critical role of simultaneously contrasting instructions and passages: instruction contrasts enable the model to understand their guiding function, while passage contrasts reinforce the alignment between instruction-aware queries and relevant passages.

• **Univariate vs. Multivariate Loss.** Empirical results demonstrate clear advantages of multivariate contrastive objectives over simpler univariate objectives, including those used by Promptriever (Weller et al., 2025). The superior performance emphasizes the importance of jointly modeling the interactions among instructions, queries, and passages.

Table 6: Detailed configuration comparison on Follow-IR (Weller et al., 2024) benchmarking multiple loss function designs and varying sizes of backbone LMs.

**Base Model (→): ModernBERT | Qwen2.5-1.5B**

| Dataset (→) | Robust04 | | News21 | | Core17 | | Overall | | Robust04 | | News21 | | Core17 | | Overall | |
|---|---|---|---|---|---|---|---|---|---|---|---|---|---|---|---|---|
| Config. (↓) | MAP | p-MRR | nDCG | p-MRR | MAP | p-MRR | score | p-MRR | MAP | p-MRR | nDCG | p-MRR | MAP | p-MRR | score | p-MRR |
| Base | 4.29 | -5.75 | 4.27 | -1.44 | 5.73 | -0.53 | 4.76 | -0.27 | 4.71 | -0.51 | 7.46 | -0.16 | 5.91 | 1.56 | 6.03 | 0.29 |
| w/ $\ell_P^{uni}$ | 11.25 | -1.75 | 6.80 | 0.36 | **11.04** | 0.72 | 9.70 | -0.22 | 15.52 | 1.87 | 15.08 | **3.77** | 12.23 | 1.16 | 14.28 | 2.27 |
| w/ $\ell_I^{uni}$ | 3.80 | -1.58 | 1.01 | -0.51 | 5.94 | -0.91 | 3.58 | -1.00 | 8.77 | -1.54 | 11.54 | 0.76 | 7.55 | 0.26 | 9.29 | -0.18 |
| w/ $\ell_{IQ}^{uni}$ | 10.40 | -2.04 | 7.55 | 0.62 | 10.22 | 1.48 | 9.39 | 0.02 | 14.89 | 2.45 | 12.2 | 1.09 | 11.73 | 1.53 | 12.94 | 1.69 |
| w/ $\ell_{P,I}^{uni}$ | 9.56 | -0.04 | 6.10 | 1.40 | 9.09 | 1.88 | 8.25 | 1.08 | 15.93 | 1.74 | 13.99 | 1.57 | **13.22** | **2.69** | 14.38 | 2.00 |
| w/ $\ell_{P,IQ}^{uni}$ | 10.77 | -1.31 | 6.63 | 0.31 | 10.51 | 0.92 | 9.30 | -0.03 | 15.62 | 1.22 | 12.81 | 1.45 | 12.48 | 1.69 | 13.64 | 1.46 |
| w/ $\ell_{I,IQ}^{uni}$ | 8.41 | -0.78 | 5.03 | 0.58 | 9.09 | 1.49 | 7.51 | 0.43 | 15.19 | 0.83 | 13.06 | 1.26 | 11.17 | 1.62 | 13.14 | 1.24 |
| w/ $\ell_{P,I,IQ}^{uni}$ | 9.74 | -0.39 | 6.38 | 0.11 | 9.71 | 2.81 | 8.61 | 0.84 | 15.22 | 1.09 | 14.06 | 1.46 | 12.20 | 1.93 | 13.83 | 1.49 |
| w/ $\ell_{P,I}^{multi}$ | **13.81** | -0.36 | **14.97** | -1.05 | 11.04 | 1.67 | **13.27** | 0.09 | 16.77 | **4.95** | 14.11 | 2.71 | 12.68 | 1.95 | **14.52** | **3.20** |
| w/ $\ell_{P,IQ}^{multi}$ | 10.36 | -0.92 | 6.36 | 0.39 | 10.42 | 1.41 | 9.05 | 0.30 | 15.69 | 3.32 | 11.85 | 1.43 | 11.99 | 1.47 | 13.18 | 2.07 |
| w/ $\ell_{I,IQ}^{multi}$ | 9.10 | -1.97 | 6.83 | **0.90** | 9.15 | 0.72 | 8.36 | -0.12 | 15.18 | 1.51 | **15.34** | 0.62 | 11.68 | 1.13 | 14.07 | 1.09 |
| w/ $\ell_{P,I,IQ}^{multi}$ | 9.96 | **0.28** | 5.99 | 0.06 | 9.79 | **2.92** | 8.58 | **1.09** | 14.13 | 1.72 | 12.57 | 1.16 | 11.78 | 2.05 | 12.83 | 1.65 |

**Base Model (→): Qwen2.5-1.5B-Instruct | Llama3.2-1B**

| Dataset (→) | Robust04 | | News21 | | Core17 | | Overall | | Robust04 | | News21 | | Core17 | | Overall | |
|---|---|---|---|---|---|---|---|---|---|---|---|---|---|---|---|---|
| Config. (↓) | MAP | p-MRR | nDCG | p-MRR | MAP | p-MRR | score | p-MRR | MAP | p-MRR | nDCG | p-MRR | MAP | p-MRR | score | p-MRR |
| Base | 4.73 | -1.19 | 9.82 | 2.30 | 6.40 | 0.78 | 6.98 | 0.63 | 8.04 | -1.48 | 17.70 | 1.50 | 9.76 | 0.42 | 11.81 | 0.14 |
| w/ $\ell_P^{uni}$ | **17.91** | 3.86 | **17.52** | 0.65 | **13.59** | 3.79 | **16.34** | 2.76 | 16.75 | **5.98** | 20.80 | 0.65 | 13.89 | **3.81** | 17.15 | **3.48** |
| w/ $\ell_I^{uni}$ | 8.52 | -1.43 | 9.20 | -1.31 | 7.30 | 0.54 | 8.34 | -0.73 | 7.60 | -3.00 | 12.58 | -1.39 | 7.32 | 2.94 | 9.17 | -0.48 |
| w/ $\ell_{IQ}^{uni}$ | 15.07 | 3.36 | 14.97 | 1.68 | 12.06 | 0.90 | 14.03 | 1.98 | 17.18 | 1.76 | 24.80 | 0.46 | 14.10 | 3.25 | 18.69 | 1.82 |
| w/ $\ell_{P,I}^{uni}$ | 15.53 | 2.35 | 17.27 | 0.11 | 12.55 | 2.36 | 15.12 | 1.61 | 17.11 | 0.88 | 23.49 | 2.24 | 13.33 | 3.21 | 17.98 | 2.11 |
| w/ $\ell_{P,IQ}^{uni}$ | 16.30 | 3.71 | 13.83 | 1.56 | 12.03 | 2.60 | 14.05 | 2.62 | 16.15 | -0.35 | 23.47 | 1.65 | 12.06 | 2.05 | 17.23 | 1.12 |
| w/ $\ell_{I,IQ}^{uni}$ | 14.91 | 2.52 | 13.62 | 0.72 | 11.28 | 1.73 | 13.27 | 1.66 | 18.08 | -1.27 | **26.11** | 2.04 | 13.18 | 3.30 | 19.12 | 1.36 |
| w/ $\ell_{P,I,IQ}^{uni}$ | 16.25 | 3.10 | 13.72 | 0.86 | 12.35 | **3.94** | 14.11 | 2.64 | 18.06 | -0.38 | 22.33 | 1.42 | 13.50 | 3.23 | 17.96 | 1.42 |
| w/ $\ell_{P,I}^{multi}$ | 16.62 | **4.18** | 16.33 | -0.13 | 12.59 | 3.49 | 15.18 | 2.51 | 19.19 | 0.89 | 23.65 | **3.02** | 14.33 | 2.99 | 19.05 | 2.30 |
| w/ $\ell_{P,IQ}^{multi}$ | 15.67 | 2.16 | 14.29 | **2.65** | 12.22 | 3.13 | 14.06 | 2.65 | 16.02 | 3.03 | 23.83 | 1.05 | 12.85 | 1.92 | 17.57 | 2.00 |
| w/ $\ell_{I,IQ}^{multi}$ | 14.95 | 3.90 | 13.76 | 1.36 | 12.13 | 1.34 | 13.61 | 2.20 | **19.36** | -14.11 | 23.61 | 1.34 | **14.67** | 0.91 | **19.21** | 0.38 |
| w/ $\ell_{P,I,IQ}^{multi}$ | 15.80 | 2.72 | 15.03 | 2.10 | 11.84 | 3.13 | 14.22 | 2.65 | 17.11 | 1.24 | 25.09 | 2.49 | 14.06 | 2.62 | 18.75 | 2.11 |

**Base Model (→): Llama3.2-1B-Instruct | Qwen2.5-3B**

| Dataset (→) | Robust04 | | News21 | | Core17 | | Overall | | Robust04 | | News21 | | Core17 | | Overall | |
|---|---|---|---|---|---|---|---|---|---|---|---|---|---|---|---|---|
| Config. (↓) | MAP | p-MRR | nDCG | p-MRR | MAP | p-MRR | score | p-MRR | MAP | p-MRR | nDCG | p-MRR | MAP | p-MRR | score | p-MRR |
| Base | 8.60 | -2.07 | 11.05 | 0.63 | 8.72 | 0.15 | 9.45 | -0.43 | 4.97 | -0.82 | 8.27 | 0.83 | 5.76 | 1.14 | 6.33 | 0.38 |
| w/ $\ell_P^{uni}$ | 18.41 | **6.30** | 26.63 | 2.09 | 14.37 | 2.89 | 19.81 | 3.76 | 17.57 | 3.00 | 19.16 | -1.53 | 12.66 | 1.90 | **16.46** | 1.12 |
| w/ $\ell_I^{uni}$ | 9.11 | -2.05 | 13.43 | 1.04 | 8.62 | 0.71 | 10.39 | -0.10 | 7.75 | -2.76 | 9.33 | -1.61 | 6.66 | -0.91 | 7.91 | -1.76 |
| w/ $\ell_{IQ}^{uni}$ | 17.63 | 3.00 | 22.32 | **3.89** | 12.81 | 2.03 | 17.59 | 2.98 | 15.35 | 0.41 | 16.62 | 0.49 | 11.89 | 1.22 | 14.62 | 0.71 |
| w/ $\ell_{P,I}^{uni}$ | 18.01 | 4.52 | 25.98 | 1.90 | 14.79 | 2.24 | 19.59 | 2.89 | 16.64 | 4.52 | 17.13 | 1.26 | 12.24 | 2.82 | 15.34 | 2.87 |
| w/ $\ell_{P,IQ}^{uni}$ | 17.23 | 3.65 | 24.20 | 2.89 | 14.02 | 3.08 | 18.48 | 3.21 | 17.44 | 3.05 | 16.81 | 1.34 | 12.45 | 1.84 | 15.57 | 2.08 |
| w/ $\ell_{I,IQ}^{uni}$ | 17.73 | -0.59 | 24.58 | 2.77 | 14.68 | 2.33 | 19.00 | 1.50 | 16.61 | 0.00 | 16.23 | 0.89 | 12.25 | 1.73 | 15.03 | 0.87 |
| w/ $\ell_{P,I,IQ}^{uni}$ | 18.88 | -0.19 | 24.79 | 3.67 | **16.27** | **4.25** | 19.98 | 2.58 | 17.67 | **4.58** | 17.31 | 0.79 | **13.44** | 2.67 | 16.14 | 2.68 |
| w/ $\ell_{P,I}^{multi}$ | **19.12** | 5.58 | 26.12 | 3.80 | 15.22 | 1.90 | **20.15** | **3.76** | 17.58 | 4.28 | **19.48** | 1.09 | 12.18 | **3.62** | 16.41 | **3.00** |
| w/ $\ell_{P,IQ}^{multi}$ | 18.12 | 4.42 | 21.80 | 2.82 | 13.20 | 3.23 | 17.71 | 3.49 | 15.60 | 1.93 | 19.11 | 1.41 | 11.76 | 1.14 | 15.49 | 1.49 |
| w/ $\ell_{I,IQ}^{multi}$ | 19.03 | -0.99 | 24.23 | 2.44 | 14.40 | 3.43 | 19.22 | 1.63 | 15.45 | 0.33 | 15.68 | **1.81** | 10.86 | 1.82 | 14.00 | 1.32 |
| w/ $\ell_{P,I,IQ}^{multi}$ | 17.88 | 2.51 | **26.86** | 1.77 | 14.55 | 2.65 | 19.76 | 2.31 | **17.72** | 2.92 | 18.35 | 0.37 | 12.67 | 1.03 | 16.25 | 1.44 |

**Base Model (→): Qwen2.5-3B-Instruct | Average**

| Dataset (→) | Robust04 | | News21 | | Core17 | | Overall | | Robust04 | | News21 | | Core17 | | Overall | |
|---|---|---|---|---|---|---|---|---|---|---|---|---|---|---|---|---|
| Config. (↓) | MAP | p-MRR | nDCG | p-MRR | MAP | p-MRR | score | p-MRR | MAP | p-MRR | nDCG | p-MRR | MAP | p-MRR | score | p-MRR |
| Base | 4.95 | -1.30 | 9.68 | 2.42 | 6.58 | -0.41 | 7.07 | 0.24 | 5.76 | -1.87 | 9.75 | 0.87 | 6.98 | 0.44 | 7.49 | 0.14 |
| w/ $\ell_P^{uni}$ | 17.65 | **4.29** | 18.59 | 1.10 | 13.59 | 2.00 | 16.61 | 2.46 | 16.44 | **3.36** | 17.80 | 1.01 | 13.05 | 2.32 | 15.76 | 2.23 |
| w/ $\ell_I^{uni}$ | 7.76 | -1.57 | 9.64 | -0.93 | 6.60 | -0.54 | 8.00 | -1.01 | 7.62 | -1.99 | 9.53 | -0.56 | 7.14 | 0.30 | 8.10 | -0.75 |
| w/ $\ell_{IQ}^{uni}$ | 19.67 | 1.83 | 20.65 | 1.58 | 12.05 | -0.82 | 17.45 | 0.86 | 15.74 | 1.54 | 17.02 | 1.40 | 12.12 | 1.37 | 14.96 | 1.44 |
| w/ $\ell_{P,I}^{uni}$ | 19.69 | 2.02 | 20.90 | -0.26 | **14.61** | 2.06 | 18.40 | 1.27 | 16.07 | 2.28 | 17.84 | 1.17 | 12.83 | 2.47 | 15.58 | 1.98 |
| w/ $\ell_{P,IQ}^{uni}$ | 19.08 | 3.87 | 17.96 | 0.54 | 13.16 | 1.98 | 16.73 | 2.13 | 16.08 | 1.98 | 16.53 | 1.39 | 12.39 | 2.02 | 15.00 | 1.80 |
| w/ $\ell_{I,IQ}^{uni}$ | 19.14 | -1.68 | 15.65 | 0.91 | 12.03 | 0.52 | 15.61 | -0.08 | 15.72 | -0.14 | 16.33 | 1.31 | 11.95 | 1.82 | 14.67 | 1.00 |
| w/ $\ell_{P,I,IQ}^{uni}$ | 19.66 | 1.92 | 18.79 | 1.17 | 13.46 | 0.70 | 17.30 | 1.26 | 16.50 | 1.39 | 16.77 | 1.35 | 12.99 | **2.79** | 15.42 | 1.84 |
| w/ $\ell_{P,I}^{multi}$ | 19.63 | 3.31 | **22.41** | 1.82 | 14.58 | **3.69** | 18.87 | 2.94 | **17.53** | 3.26 | **19.58** | 1.61 | **13.23** | 2.76 | **16.78** | **2.54** |
| w/ $\ell_{P,IQ}^{multi}$ | 19.98 | 3.37 | 19.04 | 1.68 | 13.50 | 1.91 | 17.50 | 2.32 | 15.92 | 2.47 | 16.61 | **1.63** | 12.28 | 2.03 | 14.94 | 2.05 |
| w/ $\ell_{I,IQ}^{multi}$ | 17.83 | 1.26 | 17.24 | 1.46 | 13.01 | 0.90 | 16.03 | 1.20 | 15.84 | -1.44 | 16.67 | 1.42 | 12.27 | 1.46 | 14.93 | 1.10 |
| w/ $\ell_{P,I,IQ}^{multi}$ | **20.05** | 2.98 | 18.90 | **1.95** | 13.92 | 1.82 | 17.62 | 2.25 | 16.09 | 2.05 | 17.54 | 1.41 | 12.66 | 2.32 | 15.43 | 1.93 |

• **Encoder-Only vs. Decoder-Only Models.** Our experiments reveal that decoder-only models consistently outperform encoder-only models in retrieval effectiveness and instruction-following tasks. We attribute this improvement primarily to the increased parameter capacity and extensive pre-training data utilized in large language model training phases.

## G.3    COMPARISON WITH RERANKING BASELINES

Retrieval and reranking have different, sequential roles. The first-stage retriever searches a large corpus under strict latency and memory limits to return a compact candidate set. A reranker then

Table 7: Comparison with reranking baselines on FollowIR (Weller et al., 2024) dataset.

| Model/p-MRR | Robust04 | News21 | Core17 | FollowIR |
|---|---|---|---|---|
| InF-Embed (e5-base-v2) | 6.9 | 3.2 | 5.3 | 5.1 |
| InF-Embed (e5-large-v2) | 9.4 | 2.0 | 7.1 | 6.2 |
| InF-Embed (Llama-3.2-1B) | 6.0 | 0.7 | 3.8 | 3.5 |
| InF-Embed (Qwen2.5-1.5B) | 4.9 | 2.7 | 1.9 | 3.2 |
| InF-Embed (e5-mistral) | 6.2 | 1.5 | 6.3 | 4.7 |
| InF-Embed (Qwen2.5-7B) | 6.4 | 1.8 | 6.5 | 4.9 |
| FLAN-T5-base | 5.3 | -0.1 | -3.3 | 0.6 |
| Llama-2-7B-chat | 2.0 | 0.2 | 2.8 | 1.7 |
| FLAN-T5-large | 3.9 | 8.9 | 1.3 | 4.7 |

Table 8: Comparison with reranking baselines on MAIR (Sun et al., 2024) dataset.

| Model/NDCG@10 | DD-15 | DD-16 | DD-17 | FR-21 | FR-22 | MAIR |
|---|---|---|---|---|---|---|
| InF-Embed (e5-base-v2) | 47.5 | 35.5 | 32.9 | 49.8 | 78.9 | 48.9 |
| InF-Embed (e5-large-v2) | 51.4 | 37.9 | 34.7 | 57.0 | 89.2 | 54.0 |
| InF-Embed (Llama-3.2-1B) | 50.5 | 36.8 | 36.7 | 57.1 | 87.0 | 53.6 |
| InF-Embed (Qwen2.5-1.5B) | 44.2 | 23.4 | 35.4 | 52.6 | 85.1 | 48.1 |
| InF-Embed (e5-mistral) | 52.0 | 37.3 | 37.4 | 58.4 | 89.1 | 54.8 |
| InF-Embed (Qwen2.5-7B) | 47.6 | 32.1 | 36.8 | 51.5 | 86.6 | 50.9 |
| Bge-reranker-v2-m3 | 53.4 | 35.7 | 42.3 | 45.4 | 85.7 | 52.5 |
| Bge-reranker-v2-gemma | 57.5 | 36.6 | 45.4 | 50.2 | 80.1 | 53.9 |
| Mxbai-rerank-large-v1 | 49.2 | 29.4 | 37.9 | 18.5 | 66.4 | 40.3 |

reorders that set using more expressive but slower models. As a result, rerankers usually achieve higher accuracy than standalone retrievers, but at higher computational cost and serving latency.

To compare fairly, we evaluate instruction-aware rerankers on FollowIR and MAIR using the same inputs and candidate pools. For each query, we prompt the reranker with the instruction, the query, and each candidate passage to obtain a relevance score, and then reorder the candidates. As shown in Table 7 and Table 8, our InF-Embed retriever, which performs single-pass embedding retrieval, matches or surpasses these rerankers while using far fewer compute-intensive operations. Adding an instruction-aware reranker on top of InF-Embed yields further gains, indicating that InF-Embed provides a strong first-stage representation for instruction-following search.

### G.4 BROADER APPLICATIONS

Table 9: Model performance (NDCG@10) on broader applications in personalization.

| Models | PointRec | CPCD |
|---|---|---|
| E5-base-v2 | 40.75 | 1.90 |
| Inf-Embed (E5-base-v2) | 46.22 | 2.23 |
| E5-large-v2 | 40.37 | 3.70 |
| Inf-Embed (E5-large-v2) | 46.63 | 3.91 |

We further test InF-Embed on two personalization tasks: (1) PointRec (Afzali et al., 2021), a benchmark for narrative-driven point-of-interest recommendation where instructions describe a user's situational needs; and (2) CPCD (Chaganty et al., 2023), a dataset for conversational playlist curation that models preferences over sets of items. The results in Table 9 show consistent gains in NDCG@10 when training with InF-Embed, suggesting that instruction-aware representation learning is also useful for downstream recommendation tasks without modifying the task model.

## H   PROMPT DETAILS

We include query-synthesis prompt details as follows:

---

**Query Synthesis Prompt – Part I**

```
You are given a document along with a search query and an instruction
    that retrieves this document.

Document: {document}
Positive Query: {query_positive}
Positive Instruction: {instruction_positive}

Your task is to generate a NEW search query that will lead to the
    creation of DISTINCTLY DIFFERENT documents. The new query combined
    with the original instruction needs to create documents that are
    easily distinguishable from the original document when evaluated.
```

---

**Query Synthesis Prompt – Part II**

```
To create effective negative examples:
1. IDENTIFY KEY ELEMENTS: First, identify 2-3 core aspects/facts/claims
    of the original document.
2. CREATE SEMANTIC OPPOSITES:
   - Your new query should target information that contradicts or
       significantly diverges from these core aspects
3. MAINTAIN DOMAIN RELEVANCE: Stay in a similar subject area but with
    crucial differences:
   - Change time periods, locations, entities, or outcomes
   - Reverse cause-effect relationships
   - Switch perspective (e.g., benefits vs. drawbacks, support vs.
       opposition)
   - Modify the granularity or specificity level
4. ENSURE CLEAR DISTINCTION: A human evaluator should be able to easily
    determine which document is the original vs. synthetic based on
    these key distinctions.

The goal is that when your NEW query is used with the ORIGINAL
    instruction, they should produce documents that are clearly
    distinguishable from the original document (at least 3 significant
    differences).

Please provide your answer in the following format:
Query: <your new query>
Be concise but specific enough to ensure clear differentiation.
```

We include instruction-synthesis prompt details as follows:

**Instruction Synthesis Prompt – Part I**

```
You are given a document along with a search query and an instruction
    that retrieves this document.

Document: {document}

Positive Query: {query_positive}

Positive Instruction: {instruction_positive}

Your task is to generate a NEW instruction that will lead to the
    creation of DISTINCTLY DIFFERENT documents. The new instruction
    combined with the original query needs to create documents that are
     easily distinguishable from the original document when evaluated.

To create effective negative examples:

1. IDENTIFY KEY ELEMENTS: First, identify 2-3 core aspects/facts/claims
     of the original document.

2. CREATE SEMANTIC OPPOSITES:
  - Your new instruction should target information that contradicts or
      significantly diverges from these core aspects
```

**Instruction Synthesis Prompt – Part II**

```
3. MAINTAIN DOMAIN RELEVANCE: Stay in a similar subject area but with
    crucial differences:
  - Change time periods, locations, entities, or outcomes
  - Reverse cause-effect relationships
  - Switch perspective (e.g., benefits vs. drawbacks, support vs.
      opposition)
  - Modify the granularity or specificity level

4. ENSURE CLEAR DISTINCTION: A human evaluator should be able to easily
     determine which document is the original vs. synthetic based on
    these key distinctions.

The goal is that when your NEW instruction is used with the ORIGINAL
    query, they should produce documents that are clearly
    distinguishable from the original document (at least 3 significant
    differences).

Please provide your answer in the following format:

Instruction: <your new instruction>

Be concise but specific enough to ensure clear differentiation.
```

