# OpenReview forum: "Towards Better Instruction Following Retrieval Models"
_ICLR.cc/2026/Conference — Submitted to ICLR 2026_

### Official Review · Reviewer_xbxC · 2025-10-23

**Soundness:** 2
**Presentation:** 3
**Contribution:** 1
**Rating:** 2
**Confidence:** 5

**Summary:**

This paper presents InF-IR, a new dataset and training framework for instruction-following retrieval models. The authors aim to improve the alignment between natural-language instructions and retrieval behavior by generating synthetic <instruction, query, document> triplets. They further introduce an instruction–query attention mechanism and a marginal negative sampling strategy. Experiments on BEIR, BRIGHT, and FollowIR benchmarks show that their proposed InF-Embed model achieves higher p-MRR and nDCG compared to existing embedding models such as E5 and ModernBERT.

While the paper addresses a meaningful problem — improving retrievers’ ability to follow human-written instructions — its core contribution overlaps substantially with prior work in the community and therefore makes only limited contributions to the IR field. Moreover, the reported results show clear inconsistencies with other published works, and the paper lacks dataset examples or sufficient methodological transparency. In its current form, the paper would require major revisions and thorough re-evaluation.

**Strengths:**

1. Timely and Relevant Topic:
The paper tackles an increasingly important challenge — aligning retrievers with natural-language instructions — a direction that bridges retrieval and instruction tuning in large language models.
2. Multi-Benchmark Evaluation:
The authors evaluate their approach on BEIR, BRIGHT, and FollowIR, aiming to demonstrate its generalization ability across both reasoning-intensive and traditional retrieval benchmarks.
3. Readable and Well-Structured Writing:
The paper is overall well-written and easy to follow, with clear motivation and well-organized presentation. Figures and tables are neatly formatted, though a few inconsistencies remain (as noted later).

**Weaknesses:**

1. Limited Contribution and Lack of Depth in Instruction Analysis:
This work shows a high degree of overlap with prior instruction-following retrieval studies such as Promptriever and InfoSearch (ICLR 2025). The proposed methods appear largely incremental rather than conceptually novel. Although the paper claims to move “towards better instruction following,” it does not provide any fine-grained analysis or understanding of instructions.
For example, InfoSearch (ICLR 2025) explicitly categorizes instructions by audience, language, source, and length — dimensions crucial to understanding instruction-following behavior. In contrast, this paper treats all documents with a uniform prompt without differentiating instruction types. Moreover, the paper provides no concrete examples of the generated dataset, which seriously limits the transparency and reliability of the proposed InF-IR data.
2. Incomplete and Outdated Baselines:
The baseline comparisons are insufficient. The paper omits several direct counterparts designed for instruction-following retrieval, such as Promptriever and Follower. In addition, many of the compared baselines are outdated — for instance, more recent and stronger embedding models like Qwen3-Embeddings are not included.
The paper also lacks comparison with advanced large language models (e.g., GPT-4o), which often lead the progress in instruction-following capabilities and could serve as valuable reference points for assessing retriever alignment.
3. Unreliable Experimental Results:
The reported results contain clear anomalies. In Table 2, the nDCG@10 scores on the BRIGHT dataset (≈ 3–10 for E5 and InF-Embed) are an order of magnitude lower than expected — even simple BM25 baselines typically achieve around 13–15 in published works. This suggests potential issues in the evaluation pipeline, such as metric mis-scaling, incorrect label mapping, or partial dataset evaluation.
4. Unclear Dataset Validation Process:
The dataset validation pipeline lacks transparency. The paper does not provide the validation prompts, human annotation guidelines, or detailed criteria used by the o3-mini model to assess the quality of generated triplets.

**Questions:**

1. About BRIGHT Evaluation:
Could you clarify the evaluation pipeline for BRIGHT? Why are the nDCG values so low (3–10), and how were relevance labels processed? Did you normalize or rescale metrics?
2. About the o3-mini Validation:
What specific criteria and prompts were used to validate GPT-4o outputs? How many generated samples were rejected? Although only 100 samples are reported for validation, more details are needed — for example, how were the three annotated scores distributed across the dataset?
3. About Instruction-Query Attention:
How exactly does your proposed mechanism differ from standard cross-attention or concatenation-based encoding in prior works like Promptriever?

---

### Official Review · Reviewer_F6Az · 2025-10-28

**Soundness:** 3
**Presentation:** 3
**Contribution:** 4
**Rating:** 6
**Confidence:** 4

**Summary:**

This paper introduces InF-IR, a large-scale dataset designed for instruction-following information retrieval (IR), and InF-Embed, a contrastive embedding model trained on InF-IR to improve instruction awareness in retrievers.  The authors identify a key gap between traditional text retrieval—focused solely on query-passage semantic similarity—and the growing need for instruction-conditioned retrieval, where the retriever must interpret and follow user-specified constraints (e.g., sentiment, style, or stance).

**Strengths:**

- High-quality dataset construction: The multi-stage synthesis + filtering pipeline results in superior data diversity and accuracy.
- Principled learning formulation: The introduction of multivariate conditional contrastive learning elegantly models instruction–query–passage dependencies.
- Comprehensive evaluation: Benchmarks span diverse instruction types, with consistent, reproducible gains.
- Clear ablations: The study effectively isolates the contributions of data filtering, objective design, and model architecture.
- Practical impact: Embedding-based retrievers achieve near-reranker performance with much lower computational cost.

**Weaknesses:**

- Synthetic bias: Heavy reliance on GPT-4o-mini for both data generation and negative synthesis may limit generalization to real-world, noisy user instructions.
- Limited cross-domain validation: All experiments are text-only; evaluating on multimodal or knowledge-intensive tasks would strengthen claims about generalizability.
- Interpretability: While performance gains are clear, the paper offers limited qualitative analysis (e.g., attention visualization or case studies) explaining how the model learns instruction semantics.

**Questions:**

1. How does InF-Embed perform on out-of-distribution instructions (e.g., compositional or multilingual inputs)?
2. The multivariate conditional contrastive objective appears central to the contribution. Could the authors provide more intuition or theoretical justification for why this formulation better captures instruction–query interactions than univariate objectives?
3. Are there any thematic or stylistic distribution biases between the instructions and text in the InF-IR dataset? Is the instruction diversity sufficient compared to real user data?

---

### Official Review · Reviewer_uKUi · 2025-10-31

**Soundness:** 1
**Presentation:** 3
**Contribution:** 2
**Rating:** 2
**Confidence:** 4

**Summary:**

The paper proposes a new dataset, INF-IR, which extends traditional retrieval datasets by incorporating instructions, in turn facilitating the training of “instruction-following” retrieval models that retrieve documents given both a user-instruction and a provided query. Using their dataset,  they train a custom model, INF-Embed and evaluate its effectiveness across a variety of retrieval benchmarks.

The core contribution of this paper, in my opinion, is the INF-IR dataset, however, it is unclear if INF-IR is a better option versus other public datasets such as the one available from the Promptriever paper [1, 2]. More details below.

[1] Weller et al., Promptriever: Instruction-Trained Retrievers Can Be Prompted Like Language Models
[2] https://huggingface.co/datasets/samaya-ai/msmarco-w-instructions

**Strengths:**

- The contribution of a training dataset is useful and the research question is important. I do believe more focus is needed on this research direction and more training data is helpful to this endeavor.
- The data curation methodology was validated with humans, making their approach trustworthy.

**Weaknesses:**

- Weak baselines: From my understanding all of the INF-Embed models in Table 2 were trained on additional data that the baselines did not have access to. The improvement of INF-embed in this case becomes obvious as it has access to additional data, telling us nothing new. Now, if the point of the table was to show that their approach can improve various models, then I believe it is critical that they include a baseline that is trained on the INF-Embed dataset, but without instructions, i.e., the exact <query, passage> pairs from INF-Embed, omitting the instructions. This will give us an idea of how much boost is achieved from the instruction training.
- Furthermore, the most crucial baseline, Promptriever, is left to the appendix (this should be on Figure 5 and Table 2) and it appears that it consistently outperforms the authors method in terms of p-MRR. It is a fair argument that differences in performances can be attributed to Promptriever being trained with more data, but this can be controlled for perhaps with subsampling, i.e., showing INF-embed performance trained w/ Promptriever data vs. INF-IR when trained with equal amounts of data, or Promptriever trained with the INF-embed data, etc.
- While these weaknesses are focused on baselines, their results show no clear evidence that using their dataset is a better option than [2], making the contribution limited.

**Questions:**

Would it possible to show INF-embed performance trained w/ Promptriever data vs. INF-IR when trained with equal amounts of data, or Promptriever trained with the INF-embed data?
Would it possible to provide the "without instructions" ablation?

---

### Official Review · Reviewer_wwwT · 2025-11-03

**Soundness:** 3
**Presentation:** 3
**Contribution:** 3
**Rating:** 6
**Confidence:** 4

**Summary:**

This paper introduces InF-IR, a training corpus of 38K instruction-query-passage triplets designed to improve instruction-following capabilities in retrieval models. The key contribution is a data synthesis pipeline that generates hard negatives by poisoning both instructions and queries, validated using o3-mini. The authors then propose InF-Embed, an instruction-aware embedding model trained with contrastive learning objectives. Experiments across FollowIR, MAIR, and Bright benchmarks show consistent improvements, with comprehensive ablations over 12 loss configurations.

**Strengths:**

The paper addresses a real gap in instruction-following IR by providing high-quality training data where existing work offers either small-scale datasets or lower-quality synthetic data. The negative sampling strategy that independently poisons instructions and queries is more comprehensive than prior work that only contrasts documents. Quality control using o3-mini with human validation is rigorous, and the experimental evaluation is thorough—testing 7 backbone models with 12 loss variants across multiple benchmarks. The finding that multivariate contrastive loss (ℓ^multi_{P,I}) consistently outperforms univariate objectives is well-demonstrated, and the practical impact is clear: small models achieve competitive performance with larger counterparts at lower computational cost.

Overall, it is good empirical work with clear practical value—the dataset will be useful to the community, and the experimental evaluation is comprehensive. However, I think this paper would be a stronger fit for ACL* venue or SIGIR.

**Weaknesses:**

The core methodology is primarily an engineering integration of existing techniques rather than a fundamental innovation. Generating hard negatives via LLM-based perturbation and quality filtering with stronger models are established practices; the extension to instruction-query-passage triplets, while useful, is incremental. The theoretical contribution is minimal—there's no analysis of why marginal sampling preserves effectiveness or formal characterization of when multivariate objectives dominate univariate ones.  Section 3's preliminaries are overly detailed for the target audience, and the cross-attention mechanism (Eq. 4) appears oversimplified compared to the concatenation baseline. The comparison with rerankers (Tables 7-8) conflates different task settings (candidate reranking vs. full corpus retrieval). Most critically, all evaluations use synthetic instructions—there's no validation that model improvements transfer to real user queries, and the lack of computational cost analysis (training time, inference speed, o3-mini annotation cost) limits practical assessment.

**Questions:**

1. Can you provide theoretical justification or empirical analysis for why marginal sampling doesn't significantly degrade performance compared to full combinatorial sampling?

2. The cross-attention mechanism underperforms concatenation across most settings—is this a fundamental limitation or a design choice (e.g., single-layer vs. multi-head)? What is the actual computational cost breakdown: o3-mini validation time per sample, end-to-end training time on 8×A100, and inference latency comparison with baselines?

3. Table 2 shows highly variable performance across benchmarks (FollowIR vs. Bright)—what causes this inconsistency, and can you characterize which instruction types your method handles well versus poorly? Finally, have you analyzed potential bias propagation from o3-mini to the trained models, especially given that quality validation uses only 100 human-annotated samples?

---

### Meta-Review · Area_Chair_BAQ7 · 2025-12-28

**Summary:**

This paper introduces InF-IR, a training corpus of 38K instruction-query-passage triplets generated via a synthetic data generation pipeline based on o3-mini, and uses it to train an instruction-following retrieval model. Models trained on Inf-IR demonstrate consistent improvements across several benchmarks.

Reviewers acknowledged that the paper is a solid empirical work with clear practical value; the dataset and data curation pipeline are useful; and the research topic is timely and relevant.

Concerns include:
- Limited novelty and insights: The core methodology is primarily an engineering integration of existing techniques rather than a fundamentally novel idea (wwwT, xbxC).
- Evaluation limitation: All evaluations rely on synthetic instructions, with no observed gains transfer to real user queries, and heavily based on o3-mini, which raises the bias issues given that the training data is also generated by the same model. Also, it does not clarify why the chosen evaluation is preferable to existing benchmarks. (wwwT, uKUi, xbxC, F6Az)
- Inconsistency in results, raising questions about reliability (xbxC).
- Lack of strong baselines (uKUi, xbxC).
- The paper's writing needs improvements. (wwwT)
Authors do not provide responses.

**Reviewer Concerns:**

No rebuttal provided

**Reviewer Scores:**

No rebuttal provided

---

### Decision · Program_Chairs · 2026-01-26

Reject